# The histone modification reader ZCWPW1 links histone methylation to PRDM9-induced double-strand break repair

Tao Huang[1,2,3,4†], Shenli Yuan[5,6†], Lei Gao[5], Mengjing Li[1,2,3,4], Xiaochen Yu[1,2,3,4], Jianhong Zhan[5,6], Yingying Yin[1,2,3,4], Chao Liu[7], Chuanxin Zhang[1,2,3,4], Gang Lu[8], Wei Li[7], Jiang Liu[5,6,9*], Zi-Jiang Chen[1,2,3,4,10*], Hongbin Liu[1,2,3,4,8*]

[1]Center for Reproductive Medicine, Cheeloo College of Medicine, Shandong University, Jinan, China; [2]National Research Center for Assisted Reproductive Technology and Reproductive Genetics, Shandong University, Jinan, China; [3]Key laboratory of Reproductive Endocrinology of Ministry of Education, Shandong University, Jinan, China; [4]Shandong Provincial Clinical Medicine Research Center for Reproductive Health, Shandong University, Jinan, China; [5]CAS Key Laboratory of Genome Sciences and Information, Collaborative Innovation Center of Genetics and Development, Beijing Institute of Genomics, Chinese Academy of Sciences, Beijing, China; [6]University of Chinese Academy of Sciences, Beijing, China; [7]State Key Laboratory of Stem Cell and Reproductive Biology, Institute of Zoology, Chinese Academy of Sciences, Beijing, China; [8]CUHK-SDU Joint Laboratory on Reproductive Genetics, School of Biomedical Sciences, Chinese University of Hong Kong, Hong Kong, China; [9]CAS Center for Excellence in Animal Evolution and Genetics, Chinese Academy of Sciences, Kunming, China; [10]Shanghai Key Laboratory for Assisted Reproduction and Reproductive Genetics, Shanghai, China

*For correspondence:
liuj@big.ac.cn (JL);
chenzijiang@hotmail.com (Z-JC);
hongbin_sduivf@aliyun.com (HL)

†These authors contributed equally to this work

Competing interests: The authors declare that no competing interests exist.

**Abstract** The histone modification writer Prdm9 has been shown to deposit H3K4me3 and H3K36me3 at future double-strand break (DSB) sites during the very early stages of meiosis, but the reader of these marks remains unclear. Here, we demonstrate that Zcwpw1 is an H3K4me3 reader that is required for DSB repair and synapsis in mouse testes. We generated H3K4me3 reader-dead Zcwpw1 mutant mice and found that their spermatocytes were arrested at the pachytene-like stage, which phenocopies the *Zcwpw1* knock–out mice. Based on various ChIP-seq and immunofluorescence analyses using several mutants, we found that Zcwpw1's occupancy on chromatin is strongly promoted by the histone-modification activity of PRDM9. Zcwpw1 localizes to DMC1-labelled hotspots in a largely Prdm9-dependent manner, where it facilitates completion of synapsis by mediating the DSB repair process. In sum, our study demonstrates the function of ZCWPW1 that acts as part of the selection system for epigenetics-based recombination hotspots in mammals.

## Introduction

Meiotic recombination ensures the faithful transmission of the genome through the pairing and segregation of homologous chromosomes, and it increases genetic diversity by disrupting linkage relationships (*Bolcun-Filas and Schimenti, 2012*; *Handel and Schimenti, 2010*). At the molecular level, meiotic recombination is initiated by the induction of programmed DSBs that are repaired by homologous recombination, leading to gene conversion and cross over formation (*Gray and Cohen, 2016*; *Hunter, 2015*; *Zickler and Kleckner, 2015*). DSB induction is a complex process, and DSB locations

are known to be marked at the very earliest stages of meiosis by trimethylation of histone H3 on lysine 4 (H3K4me3) (*Baudat et al., 2013*; *de Massy, 2013*). In mammals, this is performed by the protein PRDM9, which is expressed in the leptotene and zygotene substages (*Parvanov et al., 2017*; *Sun et al., 2015*). PRDM9 is a DNA-binding zinc finger protein, with an exceptionally long and genetically variable zinc finger domain that determines its binding specificity (for defining recombination hotspots), while its SET domain possesses histone trimethyl transferase activity, and its KRAB domain is involved in protein–protein interactions (*Grey et al., 2018*; *Paigen and Petkov, 2018*). In yeast, the histone reader Spp1 links H3K4me3 sites at promoters with the DSB formation machinery, thus promoting DSB formation (*Acquaviva et al., 2013*; *Sommermeyer et al., 2013*). In mice, although multiple studies have shown that the H3K4me3 writer Prdm9 controls the locations of DSB formation (*Baudat et al., 2010*; *Brick et al., 2012*; *Diagouraga et al., 2018*; *Grey et al., 2017*; *Myers et al., 2010*; *Parvanov et al., 2010*; *Powers et al., 2016*), much less is known about the subsequent activities of any proteins that might read these epigenetic marks and thus participate in advancing the meiotic recombination process (*Paigen and Petkov, 2018*).

DSB formation at sites defined by PRDM9 is catalyzed by an evolutionarily conserved topoisomerase-like enzyme complex consisting of the SPO11 enzyme and its binding partner TOPOVIBL (*Bergerat et al., 1997*; *Keeney et al., 1997*; *Panizza et al., 2011*; *Robert et al., 2016*; *Vrielynck et al., 2016*). SPO11-mediated cleavage results in single-strand DNA overhangs that are subsequently coated by various proteins, including DMC1 and RAD51 (*Dai et al., 2017*; *Pittman et al., 1998*; *Tarsounas et al., 1999*). The DSBs enable homology searching and alignment to occur, which in turn promote homology synapsis and DSB repair (*Inagaki et al., 2010*). A basic feature of meiosis is that DSB-mediated interactions and repair processes occur differentially between homologous nonsister chromatids, rather than between sisters, as occurs in mitotic DSB repair (*Garcia et al., 2015*; *Keeney et al., 2014*; *Lange et al., 2011*). Some DSBs are repaired in a way that generates crossovers, wherein DNA is exchanged between homologous chromosomes (*Baudat and de Massy, 2007*). The ZMM proteins (*e.g.*, TEX11, MSH4/MSH5, and RNF212) are a group of functionally related proteins known for their roles in promoting the formation of crossovers (*Edelmann et al., 1999*; *Kneitz et al., 2000*; *Lynn et al., 2007*; *Reynolds et al., 2013*; *Yang et al., 2008*).

We previously reported that the zinc finger CW-type and PWWP domain containing 1 (Zcwpw1) protein is required for meiosis prophase I in mice, and we found that *Zcwpw1* deficiency disrupted spermatogenesis in male mice but did not disrupt oogenesis in females to the same extent (*Li et al., 2019a*). Zcwpw1 is a member of the CW-domain containing protein family (*Liu et al., 2016*; *Perry, 2003*), and its zinc finger CW (zf-CW) domain has three conserved tryptophan and four conserved cysteine residues. Structural analysis has shown that human ZCWPW1 zf-CW domain is a histone modification reader (*He et al., 2010*), while chromatin pulldown analysis has confirmed that ZCWPW1 zf-CW domain recognizes H3K4me3 marks (*Hoppmann et al., 2011*). A crystal structure of the human zf-CW domain of ZCWPW1 in complex with a peptide bearing an H3K4me3 mark revealed that four amino acids – W256, E301, T302, and W303 – are primarily responsible for the binding of ZCWPW1 zf-CW domain to H3K4me3 marks (*He et al., 2010*). However, whether the H3K4me3 reading function is required for ZCWPW1's physiological role in meiosis is still unknown.

To address the physiological role of Zcwpw1's H3K4me3 reading function, we generated an H3K4me3 reader-dead *Zcwpw1* knock-in mutant mouse line (*Zcwpw1[KI/KI]* mouse). We found that spermatocytes were arrested at the pachytene-like stage, which phenocopied the defect seen in *Zcwpw1* knock–out mice thus suggesting that H3K4me3 reader function of Zcwpw1 might facilitate meiotic recombination by facilitating the DSB repair process. Mechanistically, a series of chromatin immunoprecipitation sequencing (ChIP-seq) analyses of Zcwpw1, H3K4me3, and H3K36me3 in multiple knock–out and knock–in mouse lines established that Zcwpw1 is an H3K4me3 and H3K36me3 reader that exclusively binds at genomic loci bearing Prdm9-deposited histone modifications. Zcwpw1 localizes to DMC1-labelled DSB hotspots where it can read H3K4me3 and H3K36me3 marks. Thus, beyond demonstrating that the histone modification reader protein ZCWPW1 functions in an epigenetics-based recombination hotspot selection system, this study advances our understanding of the sequence of recruitment events that are required for crossover formation during meiosis.

## Results

### The H3K4me3 reader function of ZCWPW1 is essential for meiotic recombination

Previously, we developed *Zcwpw1* knockout mice in the C57BL/6 genetic background (*Li et al., 2019a*) and found that loss of *Zcwpw1* in male mice caused a complete failure of synapsis. This failure resulted in meiotic arrest at the zygotene to pachytene stage, and this was accompanied by incomplete DSB repair and lack of crossover formation, thus leading to male infertility. In light of the known capacity of Zcwpw1 to recognize epigenetic methylation modification marks, we designed a knock-in strategy to generate a H3K4me3 reader-dead Zcwpw1mutant mouse line (*Figure 1—figure supplement 1A*). Specifically, this knock-in mutant of Zcwpw1 had three mutations– W247I/E292R/W294P– and these mutations in mouse Zcwpw1 are equivalent to the previously reported W256I, E301R, and W303P mutations in the human ZCWPW1 protein (except for T302L in humans and S293 in mice, which are not conserved) (*Figure 1—figure supplement 1B*), and all of them are known to be essential for the H3K4me3 reader function of human ZCWPW1 (*He et al., 2010*).

Western blot analysis confirmed the absence of the ZCWPW1 protein in *Zcwpw1$^{-/-}$* testes, while the Zcwpw1$^{W247I/E292R/W294P}$ variant protein was expressed at a level similar to that of the wild type (WT) protein (*Figure 1—figure supplement 2A and B*). Consistent with the western blot data, immunofluorescence staining of frozen sections from 8-week-old WT, *Zcwpw1$^{-/-}$*, and *Zcwpw1$^{KI/KI}$* mouse testes revealed that the ZCWPW1 protein was undetectable in *Zcwpw1$^{-/-}$* spermatocytes but could still be found in Zcwpw1$^{W247I/E292R/W294P}$ mutant spermatocytes (*Figure 1—figure supplement 2C*). After confirming that the Zcwpw1$^{W247I/E292R/W294P}$ mutant protein could be expressed normally in *Zcwpw1$^{KI/KI}$* mice, we prepared testis sections from 8-week-old WT, *Zcwpw1$^{-/-}$*, and the new *Zcwpw1$^{KI/KI}$* mouse line. Hematoxylin staining showed that spermatogenesis was disrupted in both the *Zcwpw1$^{-/-}$* and *Zcwpw1$^{KI/KI}$* mice. Compared with the WT mice, the seminiferous tubules of the *Zcwpw1$^{-/-}$* and *Zcwpw1$^{KI/KI}$* mice lacked post-meiotic cell types, contained apoptotic cells, or were nearly empty. Furthermore, the WT epididymides were full of sperm, but there were no obvious sperm detected in either the *Zcwpw1$^{-/-}$* or *Zcwpw1$^{KI/KI}$* samples, suggesting meiotic arrest in these mice (*Figure 1A*).

We then analyzed chromosome spreads of spermatocytes from the testes of adult mice by immunostaining for the synaptonemal complex (SC) markers SYCP1 and SYCP3 (*Figure 1B*). Immunostaining of SYCP1 and SYCP3 showed no differences among any of the genotypes with regard to leptotene-to-zygotene progression, which appeared normal in all mice. We quantified the synapsed chromosome pairs in the nuclei of WT, *Zcwpw1$^{-/-}$*, and *Zcwpw1$^{KI/KI}$* testes from 8-week-old mice. We observed 169 spermatocytes in WT testes, and 153 spermatocytes (90.5%) had all chromosome pairs fully synapsed, with only 16 spermatocytes (9.5%) exhibiting synapsis between 4 and 18 pairs of chromosomes. In contrast, among 164 spermatocytes in *Zcwpw1$^{-/-}$* testes and 158 spermatocytes in *Zcwpw1$^{KI/KI}$* testes none had complete synapsis, and we only detected around of 8 synapsed chromosome pairs in *Zcwpw1$^{-/-}$* and *Zcwpw1$^{KI/KI}$* spermatocytes (*Figure 1—figure supplement 2D*; *Li et al., 2019a*). Thus, spermatocytes lacking the H3K4me3-reader activity of the Zcwpw1 protein have severely disrupted synapsis.

Having established that Zcwpw1 facilitates the completion of synapsis during meiosis prophase I in male mice, we observed that the Zcwpw1$^{W247I/E292R/W294P}$ mutant mice exhibited the same synapsis defect as *Zcwpw1* knockout mice, suggesting that these residues are essential for the recombination-related functions of Zcwpw1. We then performed immunofluorescence staining of chromosome spreads to evaluate the recruitment of DMC1 and RAD51 to single-stranded overhang sequences in WT and *Zcwpw1$^{KI/KI}$* mice (*Figure 1C and E*). There were no differences in the numbers of DMC1 or RAD51 foci in the leptotene or zygotene stages of the two genotypes. However, analysis of WT pachytene and *Zcwpw1$^{KI/KI}$* pachytene-like spermatocytes revealed an obvious discrepancy. Decreased numbers of DMC1 and RAD51 foci were seen in the pachytene WT spermatocytes, indicating successful repair of DSBs, but the *Zcwpw1$^{KI/KI}$* pachytene-like spermatocytes retained a large number of DMC1 and RAD51 foci (*Figure 1D and F*). These results suggest that the repair of DSBs is disrupted in the absence of a functional ZCWPW1 H3K4me3 reader protein and that Zcwpw1 might facilitate meiotic DSB repair downstream of strand invasion.

Seeking to further assess the functional contributions of Zcwpw1 in meiotic recombination, we analyzed chromosome spreads of spermatocytes from the testes of adult WT and *Zcwpw1$^{KI/KI}$* mice

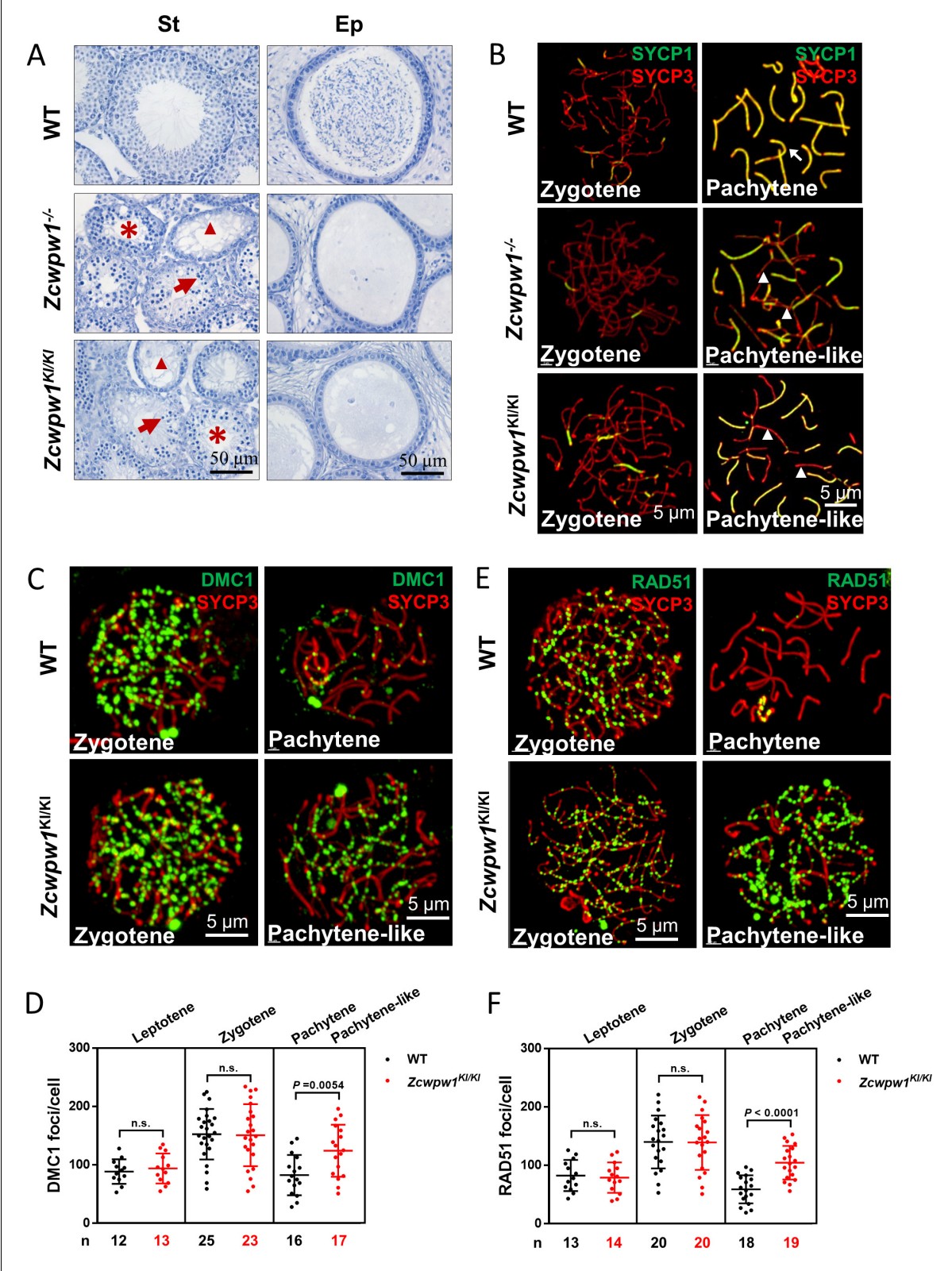

**Figure 1.** The H3K4me3 reader function of ZCWPW1 is required for synapsis and meiotic recombination. (**A**) Hematoxylin staining of adult C57BL/6 wild type, *Zcwpw1*$^{-/-}$, and *Zcwpw1*$^{KI/KI}$ testes (left panel) and epididymides (right panel). Adult *Zcwpw1*$^{-/-}$ and *Zcwpw1*$^{KI/KI}$ testis sections showed near complete arrest of spermatogenesis. Arrows, apoptotic spermatocytes; arrowheads, empty seminiferous tubules; asterisks, seminiferous tubules lacking post-meiotic spermatocytes. The spermatogenic arrest led to empty epididymides in adult *Zcwpw1*$^{-/-}$ and *Zcwpw1*$^{KI/KI}$ mice. (St) Seminiferous tubules,

*Figure 1 continued on next page*

*Figure 1 continued*

(Ep) Epididymides. Adult mice (6–8 weeks) with n = 5 for each genotype. (B) Chromosome spreads of spermatocytes from the testes of adult WT (upper panel), *Zcwpw1*⁻/⁻ (middle panel), and *Zcwpw1*^KI/KI (lower panel) males were immunostained for the SC marker proteins SYCP1 (green) and SYCP3 (red). The arrow indicates a pachytene spermatocyte in WT mice, with completely synapsed chromosomes, and the arrowheads indicate the pachytene-like spermatocytes in adult *Zcwpw1*⁻/⁻ and *Zcwpw1*^KI/KI mice with incompletely synapsed chromosomes. (C) Chromosome spreads of spermatocytes from the testes of adult WT and *Zcwpw1*^KI/KI males were immunostained for DMC1 (green) and SYCP3 (red). Representative images of spermatocytes at zygotene and pachytene in WT and at zygotene and pachytene-like stages in *Zcwpw1*^KI/KI are shown. (D) Each dot represents the number of DMC1 foci per cell, with black dots indicating WT spermatocytes and red dots indicating *Zcwpw1*^KI/KI spermatocytes. Solid lines show the mean and SD of foci number in each group of spermatocytes. P values were calculated by Student's t-test. N represents the number of cells counted, with black indicating WT spermatocytes and red indicating *Zcwpw1*^KI/KI spermatocytes. (E) Chromosome spreads of spermatocytes from the testes of adult WT and *Zcwpw1*^KI/KI males immunostained for RAD51 (green) and SYCP3 (red). (F) Each dot represents the number of RAD51 foci per cell, with black dots indicating WT spermatocytes and red dots indicating *Zcwpw1*^KI/KI spermatocytes. Solid lines show the mean and SD of the foci number for each group of spermatocytes. P values were calculated by Student's t test. Adult mice (6–8 weeks) with n = 3 for each genotype.

The online version of this article includes the following source data and figure supplement(s) for figure 1:

**Source data 1.** Number of DMC1 and RAD51 foci in *Figure 1D and F*.
**Figure supplement 1.** Generation of *Zcwpw1* reader-dead-mutant mice and *Prdm9* knockout mice.
**Figure supplement 2.** Distribution pattern of ZCWPW1 in WT, *Zcwpw1*^KI/KI and *Zcwpw1*⁻/⁻ mice.
**Figure supplement 2—source data 1.** Number of synapsed chromosome pairs per cell in *Figure 1—figure supplement 2C*.
**Figure supplement 3.** Meiotic recombination defects in *Zcwpw1* knock-in mice.
**Figure supplement 3—source data 1.** Number of MSH4 and RNF212 foci in *Figure 1—figure supplement 3B and D*.

by immunostaining for the recombination factors MSH4 and RNF212 and the Holliday junction resolution marker MLH1 (*Figure 1—figure supplement 3A and C and E*). Staining for MSH4 and RNF212 showed that the recombination machinery could assemble normally in both WT and *Zcwpw1*^KI/KI spermatocytes at the zygotene stage. However, these MSH4 and RNF212 signals decreased as expected in WT pachytene spermatocytes, but persisted on the pachytene-like *Zcwpw1*^KI/KI chromosomes (*Figure 1—figure supplement 3B and D*). Additionally, the MLH1 staining patterns indicated that Holliday junction resolution proceeded normally in mid- to late-pachytene WT spermatocytes but indicated that the recombination process was arrested in the pachytene-like spermatocytes lacking Zcwpw1 H3K4me3-reader function, which failed to progress to the pachytene stage and for which no crossover occurred, thus resulting in the absence of MLH1 foci (*Figure 1—figure supplement 3E*). These results suggest that both DSB repair and recombination are defective in the *Zcwpw1*^KI/KI mice.

To determine the specific process that can mechanistically account for the observed failure to complete meiotic recombination, we stained the spreads of spermatocytes from the testes of adult WT, *Zcwpw1*⁻/⁻, and *Zcwpw1*^KI/KI mice for the DSB marker γH2AX. We found that DSBs could form normally in all of the genotypes (*Figure 2A*), but there were obvious differences between pachytene WT spermatocytes and pachytene-like *Zcwpw1*⁻/⁻ and *Zcwpw1*^KI/KI spermatocytes. The WT pachytene spermatocytes exhibited no obvious signal for γH2AX on autosomes, but retained a γH2AX signal on the sex chromosomes, which corresponds to the XY sex body and indicates silencing of the sex chromosomes. In contrast, both autosomes and sex chromosomes retained obvious γH2AX signals, and no XY bodies were observed in *Zcwpw1*⁻/⁻ or *Zcwpw1*^KI/KI pachytene-like spermatocytes. We next stained against the DSB-repair machinery component p-ATM and found that both autosomes and sex chromosomes retained obvious p-ATM signals in the pachytene-like *Zcwpw1*⁻/⁻ and *Zcwpw1*^KI/KI spermatocytes (*Figure 2B*). Through single−stranded DNA sequencing (SSDS) by ChIP-seq against DMC1, Wells et al. found that DSBs occur in the same hotspot regions in Zcwpw1⁻/⁻ male mice (*Mahgoub et al., 2019*; *Wells et al., 2019*). Similarly, using quantitative END−seq, Mahgoub et al. also confirmed that DSBs in both WT and Zcwpw1⁻/⁻ mice completely overlapped with each other and with previously identified hotpots (*Mahgoub et al., 2019*; *Wells et al., 2019*). These results indicate that Zcwpw1 is dispensable for the induction and location of DSBs but is required for proper interhomolog interactions, including synapsis and the repair of DSBs that occur during the later steps of homologous recombination.

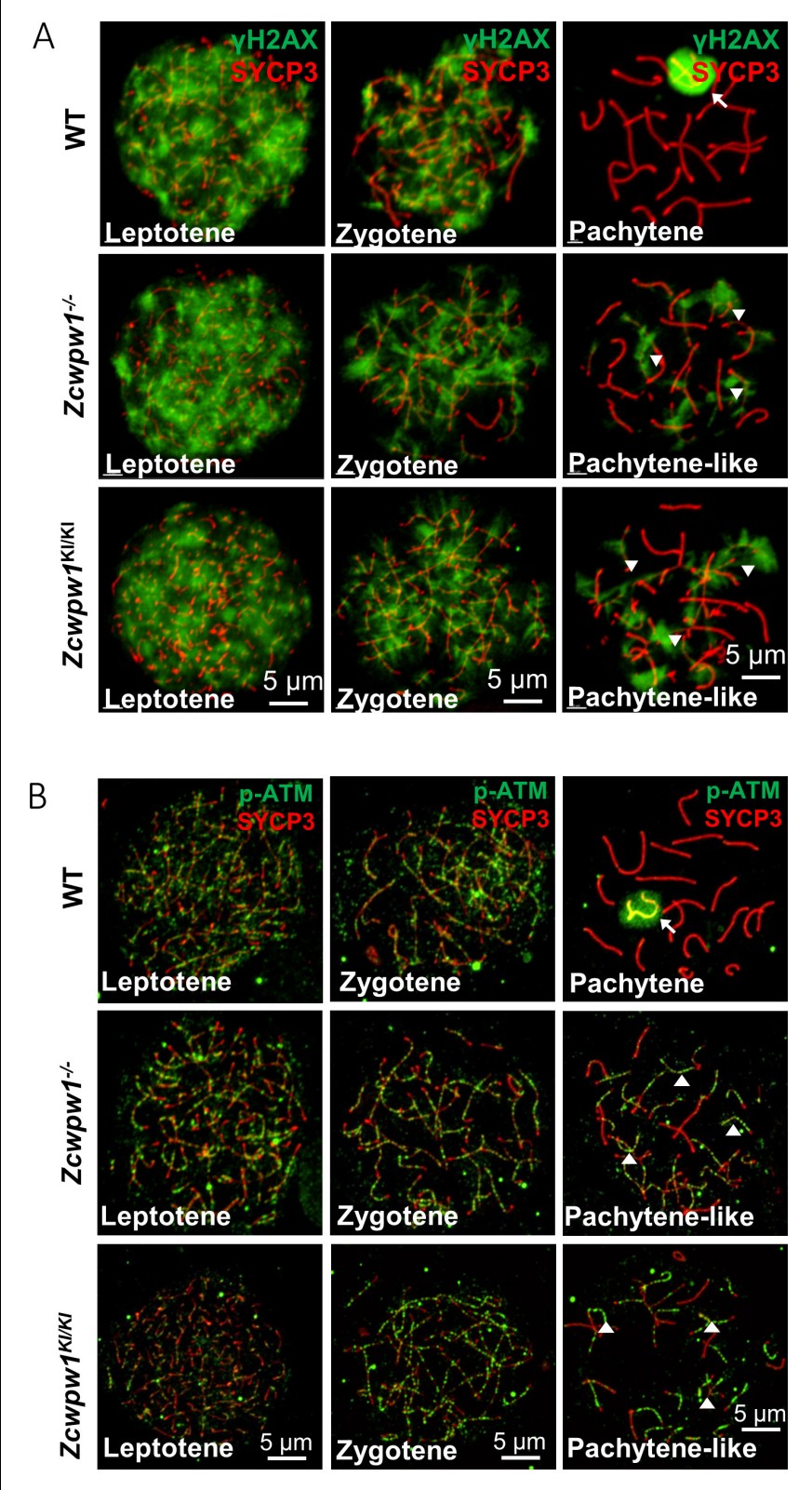

**Figure 2.** The H3K4me3 reader function of ZCWPW1 is required for DSB repair. (**A**) Chromosome spreads of spermatocytes from the testes of adult WT, *Zcwpw1*−/−, and *Zcwpw1*KI/KI males immunostained for the DSB marker γH2AX (green) and SYCP3 (red). (**B**) Chromosome spreads of spermatocytes from the testes of adult WT, *Zcwpw1*−/−, and *Zcwpw1*KI/KI males immunostained for the DSB repair protein p-ATM (green) and SYCP3 (red).
*Figure 2 continued on next page*

*Figure 2 continued*

Representative images are shown for spermatocytes at the leptotene, zygotene, pachytene (arrow indicates the XY body), and pachytene-like (arrowheads indicate the p-ATM signal) stages of the three genotypes. All experiments were performed on adult mice (6–8 weeks) with n = 3 for each genotype.

## Zcwpw1 is an H3K4me3/H3K36me3 reader

Having thus established that ZCWPW1 promotes the completion of synapsis and that it functions in meiotic recombination by facilitating DSB repair, we next investigated the mechanism by which ZCWPW1 recognizes histone modification marks involved in male meiosis prophase I. To this end, we conducted ChIP-seq using antibodies against the Zcwpw1 protein and against H3K4me3 marks. The Zcwpw1 ChIP-seq data for C57BL/6 mouse testis chromatin revealed a total of 14,688 Zcwpw1 peaks, with 499 peaks localized within 2,000 bp upstream of a transcription start site (TSS), 2146 peaks localized in exons, 6172 peaks localized in introns, and 5871 peaks localized within intergenic regions (*Figure 3—figure supplement 1A*).

Among all ZCWPW1 binding sites detected in mouse testes, 11.5% of the ZCWPW1 binding sites (peaks) overlapped with promoters, while 6.1% of ZCWPW1 binding sites overlapped with CpG islands (*Figure 3—figure supplement 1B*). Compared with the random binding sites, ZCWPW1 binding sites were not significantly enriched in the transposable element regions (*Figure 3—figure supplement 1C, D*). In HEK293T cells, Wells et al. found that a large proportion of the weakly-binding ZCWPW1 sites overlapped with Alu repeats. Notably, the weakest ZCWPW1 peaks overlapped most frequently with Alus repeats, while the strongest peaks were depleted of Alus repeats relative to random overlap. They also found that ZCWPW1 appears to have a greater affinity for methylated CpG pairs but retains some affinity even for non-methylated regions (*Mahgoub et al., 2019*; *Wells et al., 2019*). Because we found that 1766 of the ZCWPW1 binding sites overlapped with promoters, we sought to examine the transcriptome in *Zcwpw1*$^{-/-}$ testes by RNA-seq to investigate whether ZCWPW1 affected the expression of those genes whose promoters overlapped with ZCWPW1 binding sites. Analysis of RNA-seq data of postnatal day 14 (PD14) WT and *Zcwpw1*$^{-/-}$ mice identified 567 differentially expressed genes (DEGs), including 464 downregulated and 103 upregulated DEGs in *Zcwpw1*$^{-/-}$ testes compared with WT testes (*Figure 3—figure supplement 2A*). Gene ontology analysis showed that the down-regulated genes were enriched in axoneme assembly, male gamete generation and flagellated sperm motility (*Figure 3—figure supplement 2B*). However, most of the DEGs were not the genes whose promoters overlapped with ZCWPW1 binding sites (*Figure 3—figure supplement 2C*). These results strongly suggest that ZCWPW1 may not affect the transcription level of many genes sharing promoter overlap with ZCWPW1, even though *Zcwpw1*$^{-/-}$ (vs WT) spermatocytes have two up-regulated genes and 45 down-regulated genes with such promoter overlap (*Supplementary file 3*).

The H3K4me3 ChIP-seq data in C57BL/6 mice revealed a total of 55,801 H3K4me3 peaks, consistent with a previous report of 55,497 H3K4me3 peaks in whole testes (*Smagulova et al., 2011*). Lam et al. described a method for isolating pure sub-populations of meiotic substage nuclei, and they detected a total of 75,771 H3K4me3 peaks among isolated SCP3$^+$H1T$^-$ spermatocytes; this signature defines pre-leptotene to early-pachytene substage spermatocytes (*Lam et al., 2019*). In our work, we obtained a weaker average H3K4me3 signal in ZCWPW1 peaks in whole testes than that reported by Lam et al. in isolated, stage-specific spermatocyte nuclei (*Figure 3C*). The ChIP-seq data from sorted meiotic cells thus allowed the elimination of H3K4me3 peaks originating from cells that did not express ZCWPW1. In light of the known capacity of Zcwpw1 to recognize epigenetic methylation modification marks, we compared the ZCWPW1 peaks with these two sets of H3K4me3 peaks, and we found that 97.8% (14,369 of 14,688 peaks) of the Zcwpw1 peaks overlapped with the H3K4me3 peaks reported by Lam et al, while 39.4% (5,792 of 14,688 peaks) of the Zcwpw1 peaks overlapped with the H3K4me3 peaks in our data (*Figure 3A and B*), therefore supporting the hypothesis that this specific overlap with H3K4me3 peaks serves as a means for Zcwpw1 recognition of histone modification marks.

To determine whether the H3K4me3 binding ability of ZCWPW1's CW-domain is necessary for its recruitment to chromatin in vivo, we conducted an additional ZCWPW1 ChIP-seq analysis of testes samples from PD14 WT, *Zcwpw1*$^{-/-}$, and *Zcwpw1*$^{KI/KI}$ mice. The analysis indicated that no Zcwpw1

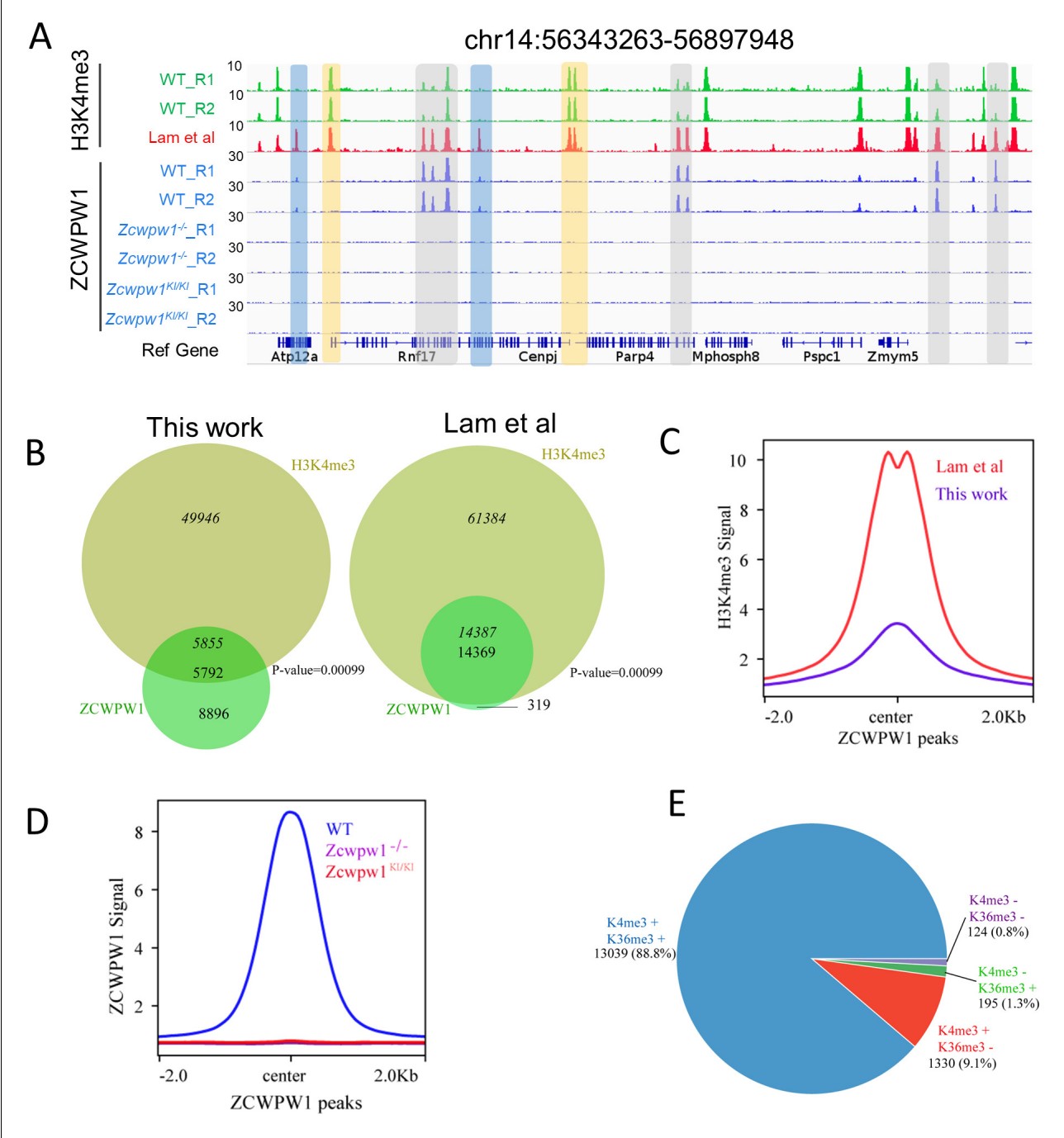

**Figure 3.** Zcpw1 is an H3K4me3 reader. (**A**) ChIP-seq genome snapshot of the distribution of H3K4me3 and ZCWPW1 peaks in C57BL/6 WT, *Zcpw1⁻/⁻*, and *Zcpw1^KI/KI* mice along a 554 kb region of chromosome 14. H3K4me3 and ZCWPW1 signals were normalized (See Methods). Overlapping regions are indicated by grey or blue shaded areas, while non-overlapping regions of interest are indicated by orange shaded areas. R1 and R2 represent two independent replicates. The H3K4me3 tract (red)was generated with isolated stage-specific (SCP3⁺H1T⁻) spermatocyte nuclei (*Lam et al., 2019*) The unit of Y axis is fold change as described in the method. (**B**) Venn diagram showing the overlap between ZCWPW1 peaks and H3K4me3 peaks. H3K4me3 data generated in whole testes (left, this study) compared with H3K4me3 data generated with isolated stage-specific (SCP3⁺H1T⁻) spermatocyte nuclei (right, Lam et al.). Italics (14,387) indicates the number of H3K4me3 peaks overlapping ZCWPW1, while standard font (14,369) indicates the number of ZCWPW1 peaks overlapping with H3K4me3 marks. Sometimes, one broad ZCWPW1 peak perhaps overlap with two narrow H3K4me3 peaks, which will cause these two numbers differ. P-values were calculated by using the permTest (see Materials and methods, ntimes = 1000). (**C**) Profile plot of averaged normalized H3K4me3 signals (see Methods) in ZCWPW1 peaks obtained in this work and in Lam et al. The profile shows the average values over 4 kb intervals for all 14,688 detected peaks (binding sites). The unit of Y axis is average fold change as described

*Figure 3 continued on next page*

*Figure 3 continued*

in the method. (D) Profile plot of the averaged ZCWPW1 signal in 14,688 ZCWPW1 peaks, in C57BL/6 WT, *Zcwpw1*$^{-/-}$ and *Zcwpw1*$^{KI/KI}$ mice. The unit of Y axis is average fold change as described in the method. (E) Pie chart showing the ratio of four ZCWPW1 peak groups determined by their overlap with histone modification peaks generated with isolated stage-specific spermatocyte nuclei (Lam et al.). The '+" indicates overlap, while '−" indicates no overlap. All ChIP-seq experiments were performed in PD14 mice with n = 4 for each genotype.

The online version of this article includes the following figure supplement(s) for figure 3:

**Figure supplement 1.** Genome-wide properties of ZCWPW1-associated binding sites.
**Figure supplement 2.** Transcriptional profiling analysis of WT and *Zcwpw1*$^{-/-}$ testes.
**Figure supplement 3.** Correlations between ZCWPW1 peaks and H3K4me3 and H3K36me3 peaks.
**Figure supplement 4.** Zcwpw1, H3K4me3 and H3K36me3 localize to the nucleus in leptotene and zygotene spermatocytes.

peaks were detected in the *Zcwpw1*$^{-/-}$ or *Zcwpw1*$^{KI/KI}$ mice (*Figure 3A and D*). These in vivo results, viewed alongside the previous reports of Zcwpw1 function in the meiotic process demonstrating that these specific mutations in the ZCWPW1 zf-CW domain affect the protein's ability to read histone modifications (including H3K4me3), together indicate that the Zcwpw1$^{W247I/E292R/W294P}$ mutant is an H3K4me3 reader-dead variant of Zcwpw1. Furthermore, these results suggest that the H3K4me3 reader function of this protein is essential for its ability to bind to chromatin and to function in meiosis prophase I in male mice.

ZCWPW1 also has a PWWP domain which was found in multiple other proteins to specifically bind to histone H3 containing an H3K36me3 mark (*Qin and Min, 2014*), so we next sought to better understand the overlap between Zcwpw1 peaks and H3K36me3 peaks in our ChIP-seq dataset. We found that 90.1% of the Zcwpw1 peaks overlapped with the H3K36me3 peaks reported by Lam et al. in isolated stage-specific spermatocyte nuclei, while 24.8% of the Zcwpw1 peaks overlapped with the H3K36me3 peaks identified by Grey et al. in whole testes (*Figure 3—figure supplement 3A*). As with our data from whole testes, the average H3K36me3 signal of ZCWPW1 peaks obtained by Grey et al. was considerably weaker than that from isolated stage-specific spermatocyte nuclei obtained by Lam et al. (*Figure 3—figure supplement 3B*).

In analyzing the correlation between ZCWPW1 binding sites and these two histone modification marks, we found that 88.8% of the ZCWPW1 peaks overlapped with regions containing both H3K4me3 and H3K36me3 marks, while only 9.1% and 1.3% of ZCWPW1 peaks overlapped with H3K4me3 and H3K36me3 peaks individually (*Figure 3E*). Furthermore, the ZCWPW1 peak intensity was significantly higher for the dual overlapping regions than for regions containing either H3K4me3 or H3K36me3 alone (*Figure 3—figure supplement 3C*). We also found that ZCWPW1 bound H3K4me3 regions had higher H3K36me3 levels than H3K4me3 regions were not bound by ZCWPW1 (*Figure 3—figure supplement 3D*). We conducted immunofluorescence analysis of chromosome spreads of spermatocytes from adult mice using rat anti-ZCWPW1 and rabbit anti-H3K4me3/H3K36me3 antibodies and found, consistent with the ChIP-seq data, that the pattern for both H3K4me3 and H3K36me3 signals were similar to the Zcwpw1 expression pattern we observed in the leptotene and zygotene stages (*Figure 3—figure supplement 4A–B*). Moreover, *Mahgoub et al., 2019* confirmed that recombinant ZCWPW1 (1–440aa) binds with the highest affinity to H3K4me3/K36me3 peptides in vitro. Taken together, these results demonstrate that ZCWPW1 preferentially binds to sites with both H3K4me3 and H3K36me3 marks.

## Zcwpw1 binding is strongly promoted by the histone modification activity of Prdm9

To identify the factors responsible for Zcwpw1 recruitment to chromatin in vivo, we searched for enriched motifs within the Zcwpw1 binding sites in our ChIP-seq data (*Figure 4—figure supplement 1A*). This analysis identified a de novo motif that is highly correlated with a known Prdm9 binding motif in mice (*Figure 4—figure supplement 1B*; *Billings et al., 2013*; *Ségurel, 2013*; *Walker et al., 2015*), and this suggested that Zcwpw1 binding to chromatin might occur in a Prdm9-dependent manner. To pursue this possibility, we compared our ZCWPW1 ChIP-seq data with previously published in vivo ChIP-seq data generated using an anti-Prdm9 antibody and with in vitro data from an affinity-seq analysis of genome-wide PRDM9 binding sites (*Grey et al., 2017*; *Walker et al., 2015*). At the genome-wide level, 13% of the Zcwpw1 peaks obtained in our study overlapped with Grey et al.'s in vivo Prdm9 peaks, while 74% of the Zcwpw1 peaks overlapped with Walker et al.'s in vitro

Prdm9 peaks. Moreover, it should be noted that 99.5% of the ZCWPW1 peaks overlapped with PRDM9 in Grey et al are covered by that ZCWPW1 peaks overlapped with PRDM9 peaks in Walker et al (*Figure 4—figure supplement 3B*). Conversely, we found that 1934 of 2,601 Prdm9 peaks (74%) from Grey et al. and 10,975 of 36,898 PRDM9 peaks (29.7%) from Walker et al. overlapped with our Zcwpw1 peaks (*Figure 4A and B*). The high overlap between Zcwpw1 and PRDM9 peaks further suggested that ZCWPW1 occupancy occurs in a PRDM9-dependent manner.

To further explore this finding of high overlap between Zcwpw1 and PRDM9 peaks in our ChIP-seq data, and in light of the well-known overlap of Prdm9 peaks with H3K4me3 and H3K36me3 marks (*Grey et al., 2017*), we compared the ZCWPW1/PRDM9 overlap with the ZCWPW1/histone mark overlap. We found that the majority of ZCWPW1 peaks overlapped with PRDM9 binding sites containing both H3K4me3 and H3K36me3 marks (*Figure 4A and C*, *Figure 4—figure supplement 1C and F*). Our further analysis of H3K4me3 peak intensity in whole testes showed that among the PRDM9-occupied regions from Grey et al., the intensity of H3K4me3 peaks overlapping with ZCWPW1 was significantly weaker than that of ZCWPW1-non-overlapping regions (*Figure 4—figure supplement 1D* left panel), which was consistent with previous reports (*Smagulova et al., 2011*). In contrast, the H3K4me3 and H3K36me3 peak intensities of isolated stage-specific spermatocyte nuclei (Lam et al.) showed that among the PRDM9-occupied regions from Grey et al. and Walker et al., the intensities of H3K4me3 and H3K36me3 peaks overlapping with ZCWPW1 were significantly greater than the intensities of ZCWPW1-non-overlapping regions (*Figure 4—figure supplement 1D and E*). Allowing for differences in the binding performance of different antibodies in different ChIP-seq analyses, the fact that some but certainly not all of the Zcwpw1 peaks overlapped with Prdm9 peaks suggests that it is the H3K4me3 and perhaps H3K36me3 epigenetic marks deposited by Prdm9, rather than the Prdm9 protein per se, that can explain the observed overlap of the Zcwpw1 and Prdm9 peaks.

To determine whether the activity of PRDM9 is necessary for ZCWPW1 recruitment to chromatin in vivo, we conducted an additional ZCWPW1 and H3K4me3 ChIP-seq analysis of testes samples from PD14 WT and *Prdm9*$^{-/-}$ mice (*Figure 1—figure supplement 1C*). Consistent with a previous report (*Brick et al., 2012*), the majority of PRDM9-dependent H3K4me3 peaks disappeared in *Prdm9*$^{-/-}$ mice (*Figure 4D and E*, *Figure 4—figure supplement 2A–C*). In our ChIP-seq data, we found that the H3K4me3 peaks overlapped with Prdm9 binding sites (*Grey et al., 2017*); with the notable exception of a 94.7% loss in Zcwpw1 binding sites, we found no obvious discrepancies between WT and *Prdm9*$^{-/-}$ testes (*Figure 4E* right panel, *Figure 4—figure supplement 2A*). This suggested that Zcwpw1 binding is strongly promoted by the specific activity of Prdm9 (*Figure 4D and E*, *Figure 4—figure supplement 2A–C*).

Having established that ZCWPW1 binding to chromatin is strongly promoted by the histone modification activity of PRDM9, we next examined changes in ZCWPW1 binding sites between WT and *Prdm9*$^{-/-}$ mutant testes. We found that although 94.7% of the ZCWPW1 peaks were apparently lost in *Prdm9*$^{-/-}$ mutant testes, 781 ZCWPW1 peaks were maintained and were accompanied by 652 newly generated ZCWPW1 peaks in *Prdm9*$^{-/-}$ mice (*Figure 4—figure supplement 3A*). Furthermore, examination of peak intensities showed that the new ZCWPW1 peaks were significantly weaker than those of both the maintained and the lost ZCWPW1 peaks in *Prdm9*$^{-/-}$ mice (*Figure 4—figure supplement 3C*). The majority of these gained (67.8%) and maintained (83.2%) ZCWPW1 peaks overlapped with promoter regions, while only 7.4% of the lost ZCWPW1 peaks overlapped with promoter regions in *Prdm9*$^{-/-}$ mice (*Figure 4—figure supplement 3D*). Further analysis showed that nearly 80% of the lost ZCWPW1 peaks overlapped with PRDM9 binding sites, while the majority of the maintained and gained ZCWPW1 peaks did not overlap with PRDM9 peaks (*Figure 4—figure supplement 3E*). Surprisingly, a motif analysis showed that 3,028 ZCWPW1 peaks, *i.e.*, those that were lost and did not overlap with PRDM9 binding sites, were significantly enriched at PRDM9 binding sites (*Figure 4—figure supplement 3F*), suggesting that Zcwpw1 binding to these sites is highly Prdm9 dependent.

## Zcwpw1 localizes to DMC1-labelled DSB hotspots in a PRDM9-dependent manner

A previous study developed a SSDS analysis using an antibody against DMC1 in mouse testes—that specifically detects protein-bound single-stranded DNA at DSB ends (*Khil et al., 2012*). SSDS provides insights into the shape and evolution of the mammalian DSB landscape (*Davies et al., 2016*).

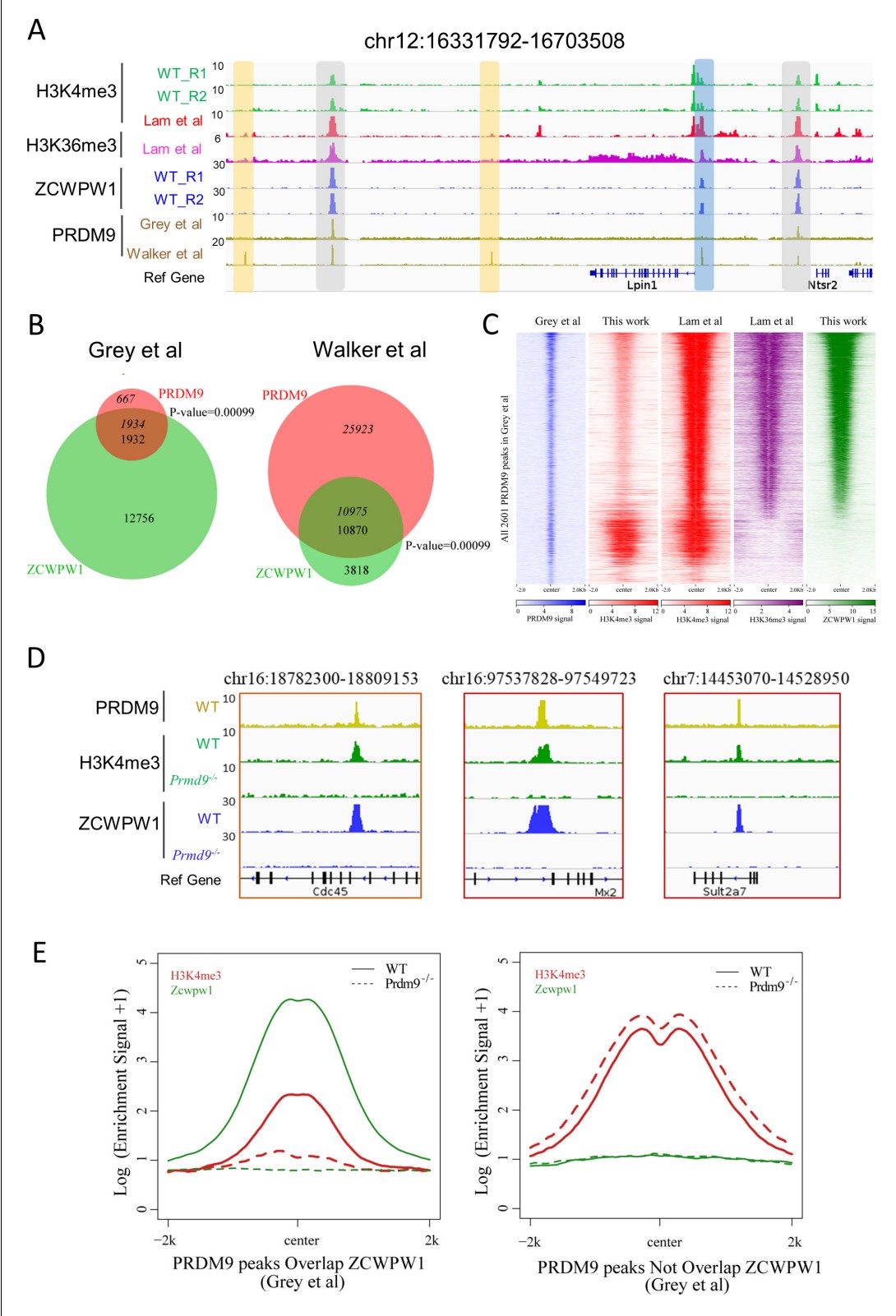

**Figure 4.** Zcwpw1 binding is strongly promoted by the histone modification activity of Prdm9. (**A**) ChIP-seq genome snapshot of the distribution of H3K4me3, H3K36me3, ZCWPW1, and PRDM9 peaks in C57BL/6 mice along a 372 kb region of chromosome 12. Overlapping peaks in samples from all four analyses are indicated by grey or blue shaded areas, while non-overlapping regions of interest are indicated by orange shaded areas. The unit of Y axis is fold change as described in the method. (**B**) Venn diagram showing the overlap between PRDM9 peaks and ZCWPW1 binding sites. On the left

*Figure 4 continued on next page*

Figure 4 continued

are in vivo PRDM9 data generated by *Grey et al., 2018*, while on the right are in vitro affinity-seq PRDM9 data generated by *Walker et al., 2015*. Italics indicate PRDM9 peak overlap with ZCWPW1, while standard font indicates ZCWPW1 peak overlap with PRDM9 peaks. (C) Heatmap showing the correlation among H3K4me3, H3K36me3, and ZCWPW1 with PRDM9 peaks (*Grey et al., 2018*). Each row represents a PRDM9 binding site of ± 2 kb around the center and ranked by ZCWPW1 signal from the highest to the lowest. Color indicates normalized ChIP-seq signal (See Methods). (D) ChIP-seq genome snapshot showing changes in H3K4me3 and ZCWPW1 binding distributions following *Prdm9* knockout (*Prdm9*$^{-/-}$) along a 27/12 kb region of chromosome 16 and a 76 kb region of chromosome 7. The PRDM9 data were obtained from *Grey et al., 2018*. The unit of Y axis is fold change as described in the method. (E) Profile plot of averaged H3K4me3 and ZCWPW1 signals obtained in this work with two types of PRDM9 peaks *Grey et al., 2018* following *Prdm9* knockout. The Y-axis shows log base-2 transformation of the normalized signal. The unit of Y axis is average fold change as described in the method. All ChIP-seq experiments were performed using PD14 mice with n = 4 for each genotype.

The online version of this article includes the following figure supplement(s) for figure 4:

**Figure supplement 1.** Correlation between Zcwpw1 binding sites and Prdm9-induced dual histone methylation.
**Figure supplement 2.** Correlation between Zcwpw1 chromatin occupancy and Prdm9-induced H3K4me3.
**Figure supplement 3.** Change in Zcwpw1 chromatin occupancy following *Prdm9* knockout.

*Lange et al., 2016* sequenced mouse SPO11 oligos and provided nucleotide-resolution DSB maps with low background and high dynamic range and found that SPO11 oligo counts correlated well with SSDS coverage. A study previously found that 94% of future DMC1 binding sites are enriched at H3K4me3 labelled hotspots, and asserted that such enrichment can be considered a global feature of DSB sites in multicellular organisms (*Smagulova et al., 2011*).Because we found that ZCWPW1 recognized dual histone modifications via PRDM9, we compared the distribution of the ZCWPW1 peaks with the DMC1 peaks and SPO11 oligos in the publicly available datasets (*Grey et al., 2017*; *Lange et al., 2016*). For the WT mice, 11,124 of the 14,688 total Zcwpw1 peaks overlapped with DMC1-defined DSB hotspots, while 10,340 of ZCWPW1 peaks overlapped with SPO11 oligo-defined DSB hotspots (*Figure 5—figure supplement 1A*). Both a heatmap and a scatter plot emphasized strong enrichment for ZCWPW1 signals at DMC1 binding sites (*Figure 5—figure supplement 1B and C*). Specifically, the greater the ZCWPW1 peak intensity, the better the overlap between the ZCWPW1 peaks and the DMC1 peaks (*Figure 5—figure supplement 1D*). These results strongly suggest that ZCWPW1 localizes to DMC1-labelled DSB hotspots.

Further analysis showed that 65.1% of the DMC1 peaks overlapped with both ZCWPW1 binding sites and merged PRDM9 peaks (*Figure 5—figure supplement 1E*). Our ZCWPW1 and H3K4me3 ChIP-seq data in WT and *Prdm9*$^{-/-}$ mice indicated an apparent lack of Zcwpw1 peaks and H3K4me3 signals at DMC1-labelled DSB hotspots (*Figure 5A and B*). These results reinforce the idea that occupancy of Zcwpw1 at DMC1-labelled DSB hotspots in *Prdm9*$^{-/-}$ spermatocytes is largely dependent on Prdm9-mediated histone modifications.

However, it bears mentioning that we also detected 781 Zcwpw1 peaks in WT testes that did not obviously overlap with DSB hotspots and we detected 652 ZCWPW1 peaks that did not obviously overlap with DSB hotspots in *Prdm9*$^{-/-}$ mice (*Figure 5B*, *Figure 4—figure supplement 3A*). We analyzed these 781 ZCWPW1 binding sites in detail and we found that 83.2% of these maintained peaks occurred within TSS ± 2 kb, a substantially larger proportion than for the average position among all lost Zcwpw1 peaks (*Figure 5—figure supplement 1F*). We also found that the distribution pattern of H3K4me3 and H3K36me3 peaks, which overlapped with those 1,433 ZCWPW1 peaks, was significantly different compared to the distribution pattern of H3K4me3 and H3K36me3 peaks that overlapped with DMC1-labelled DSB hotspots (*Figure 5B*). Thus, although it is clear that the majority of the Zcwpw1 peaks resulted from Prdm9 activity, it is possible that Zcwpw1 might have an additional transcription regulation function that is not obviously related to the Prdm9-mediated hotspot selection system.

## Discussion

Our data support a working model wherein Prdm9 binds to specific DNA motifs in the genome and writes histone modifications (H3K4me3 and H3K36me3) via the methyltransferase activity of its PR/SET domain (*Diagouraga et al., 2018*; *Powers et al., 2016*). This leads to the recruitment of proteins required for the formation of DSBs in the vicinity of its binding site (e.g., SPO11, etc.) (*Kumar et al., 2018*; *Panizza et al., 2011*; *Stanzione et al., 2016*; *Tessé et al., 2017*). After these

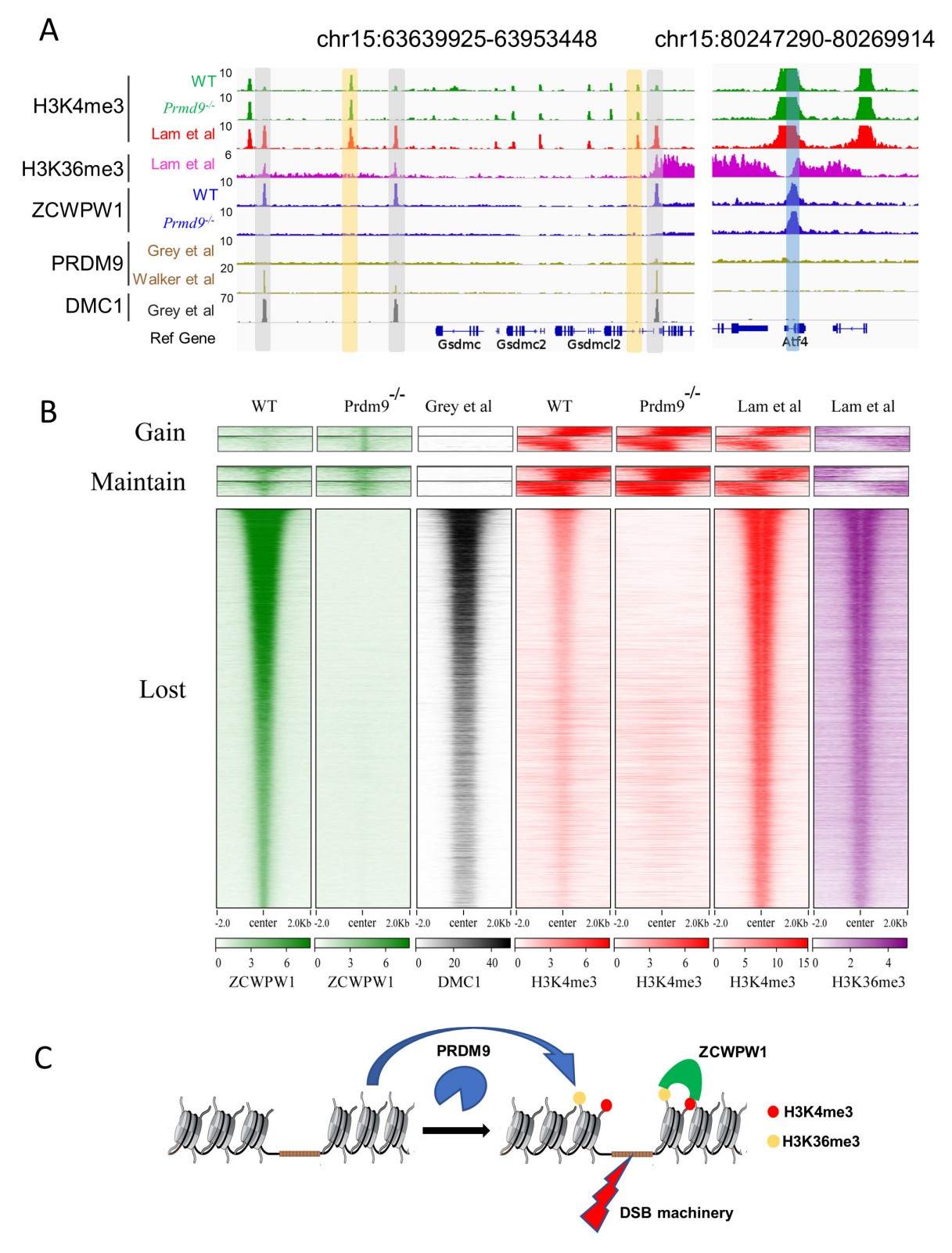

**Figure 5.** Zcwpw1 localizes to DMC1-labelled DSB hotspots in a Prdm9-dependent manner. (**A**) ChIP-seq genome snapshot of the distribution of H3K4me3, H3K36me3, ZCWPW1, PRDM9, and DMC1 peaks in C57BL/6 mice with changes in H3K4me3 and ZCWPW1 binding sites in *Prdm9* knockout mice. ZCWPW1 and DMC1 overlapping regions are indicated by gray-shaded areas, while non-overlapping regions of interest are indicated by orange or blue shaded areas. The unit of Y axis is fold change as described in the method. (**B**) Heatmap showing DMC1 (Grey et al), H3K4me3 (Lam et al),

*Figure 5 continued on next page*

Figure 5 continued

H3K36me3 (Lam et al), H3K4me3 (WT and *Prdm9*<sup>−/−</sup>), and ZCWPW1 (WT and *Prdm9*<sup>−/−</sup>) corresponding with Gained, Lost, or Maintained groups of ZCWPW1 peaks (also see *Figure 4—figure supplement 3A*). We used k-means clustering to define the 'gain' and 'maintain' group. We performed k-means clustering analysis to define the 'gain' and 'maintain' groups, and found two different subtypes within each group: a subtype with H3K4me3 signal similar along the ZCWPW1 peaks ± 2 kb, and a subtype with H3K4me3 mostly enriched on the ZCWPW1 peak center. All ChIP-seq experiments were performed using PD14 mice with n = 4 for each genotype. (C) Working model of ZCWPW1 in recognizing H3K4me3 and H3K36me3 deposited by PRDM9.

The online version of this article includes the following figure supplement(s) for figure 5:

**Figure supplement 1.** Correlation between the chromatin occupancy of Zcwpw1 and DMC1.

Prdm9-catalyzed epigenetic modifications are deposited, Zcwpw1 can specifically read these H3K4me3 and H3K36me3 marks in the vicinity of DSB sites, where Zcwpw1 functions to promote DSB repair; Zcwpw1 is not however required for DSB activity and localization (*Mahgoub et al., 2019*; *Wells et al., 2019*). This DSB-repair-promoting function greatly increases the overall completion rates of synapsis, crossover formation, and ultimately meiotic progression (*Figure 5C*).

The identification of recombination hotspots was first made in genetically-tractable experimental organisms such as bacteriophages and fungi, but it is now apparent that such hotspots are ubiquitous and active in all organisms (*Wahls, 1998*). Higher-order chromosome architecture, which can be described using the terminology of the 'tethered-loop/axis complex' model, contributes to DSB hotspot localization (*Blat et al., 2002*), and different strategies and mechanisms for the spatial regulation of DSB formation have evolved in different species, although these have many common features (*Baudat et al., 2013*; *de Massy, 2013*). In considering the evolution of hotspot selection systems, we are interested in whether other meiotic factors might have evolved in vertebrates to link PRDM9 to the meiotic recombination machinery and/or the synaptonemal complex, which would permit direct interactions with the histone marks deposited by Prdm9 (*Paigen and Petkov, 2018*; *Tock and Henderson, 2018*).

In *Saccharomyces cerevisiae*, Spp1–whose PHD finger domain is known to read H3K4me3 marks–promotes meiotic DSB formation by interacting with the axis-bound Spo11 accessory protein Mer2 (*Acquaviva et al., 2013*; *Sommermeyer et al., 2013*). Our study in mammals supports that one or more other as-yet unknown proteins might function in a similar role during DSB formation. It is noteworthy that there is structural similarity between the zf-CW domain and the PHD finger of Spp1 that helps recognize histone H3 tails (*Adams-Cioaba and Min, 2009*). Moreover, structural analysis has indicated that the zf-CW domain of human ZCWPW1 is a histone modification reader (*He et al., 2010*), and chromatin pulldown analysis has confirmed that this domain recognizes H3K4me3 marks (*Hoppmann et al., 2011*). In the present study, we showed that Zcwpw1 can specifically read H3K4me3 and H3K36me3 marks in the vicinity of DSB sites. However, somewhat surprisingly, our subsequent experiments indicated that deficiency of Zcwpw1 did not affect the recruitment of recombination-related factors like DMC1, MSH4, or RNF212, thereby implying that there might be other unknown proteins that function to link Prdm9 to meiotic recombination machinery.

The zf-CW domain of ZCWPW1 has previously been shown to bind to H3K4me3 peptides (*He et al., 2010*), and the PWWP domain, another type of 'reader' module, has been shown to recognize H3K36me3 in both peptide and nucleosome contexts (*Eidahl et al., 2013*; *Rondelet et al., 2016*; *Vezzoli et al., 2010*). Consistent with a recently deposited pre-print at bioRxiv showing that Zcwpw1 can bind to histone H3 peptides with double H3K4me3 and H3K36me3 marks with high affinity at a 1:1 ratio in vitro (*Mahgoub et al., 2019*), we also found that Zcwpw1 localized to H3K4me3 and H3K36me3 enrichment regions in ChIP-seq analysis. Notably, most of the Zcwpw1 peaks overlapping H3K4me3 peaks disappeared in *Prdm9* knockout mice. One functional implication of our study is that it is Prdm9's histone modification activity, rather than the chromatin residence of the Prdm9 protein per se, that might account for the functional interactions of the apparently co-involved Zcwpw1 and Prdm9 proteins.

Our H3K4me3 reader-dead mutant mice showed that, upon loss of the binding affinity of the Zcwpw1 zf-CW domain for H3K4me3 marks, the Zcwpw1 protein completely lost its ability to bind chromatin, and spermatocytes in mice expressing this knock-in H3K4me3 reader-dead mutant Zcwpw1 exhibited a nearly complete failure of meiosis prophase I. We also hypothesize that this protein is unable to bind unmodified histones, that is, the mutant protein might be a histone binding-

dead variant in addition to being a reader-dead variant. However, it remains unclear whether the Zcwpw1 PWWP domain, which likely functions in reading H3K36me3 marks, and/or other regions of the Zcwpw1 protein confer similarly important functions. Indeed, in future work we plan to pursue the selective disruption of the function of particular Zcwpw1 domains in our attempts to elucidate this protein's functions in male meiosis I.

While we clearly show that Zcwpw1 greatly facilitates PRDM9-dependent DSB repair, we do not yet have strong evidence for the precise nature of its functional role. One possibility is that Zcwpw1, upon binding to PRDM9-dependent histone modification hotspots, might serve as a DSB mark, which can perhaps subsequently recruit other factors involved in DSB repair. Recent studies have reported that PRDM9 binds on both the cut and uncut template chromosomes to promote meiotic recombination (*Hinch et al., 2019*; *Li et al., 2019b*). It is also possible that Zcwpw1 might directly interact with the SC machinery by using its SCP1-like domain to tether PRDM9-bound loops to the SC in order to promote homologous DSB repair.

In summary, our study identifies Zcwpw1 as an H3K4me3 and H3K36me3 reader that promotes the repair of DNA DSBs during meiotic recombination. Complementary analyses and similar conclusions were obtained by Maghoub et al and by Wells et al, establishing that ZCWPW1 is not required for DSB activity and localization (*Mahgoub et al., 2019*; *Wells et al., 2019*), findings which do not support the previous speculation that Zcwpw1 directs the location or the formation of DSBs (*Li et al., 2019a*). In future studies, we plan to focus on additional proteins (*e.g.*, ZCWPW2, MORC3/4, etc.) that have similar functional domains as Zcwpw1 (*Liu et al., 2016*), with the aim of identifying any unknown biomolecules that might act to link PRDM9 to the DSB machinery specifically or to meiotic recombination more generally.

# Materials and methods

## Key resources table

| Reagent type (species) or resource | Designation | Source or reference | Identifiers | Additional information |
|---|---|---|---|---|
| Gene<br>*Mus musculus* | *zinc finger, CW type with PWWP domain 1(Zcwpw1)* | Mouse Genome Informatics | MGI:2685899 | |
| Gene<br>*Mus musculus* | *PR domain containing 9 (Prdm9)* | Mouse Genome Informatics | MGI:2384854 | |
| Strain, strain background<br>*Mus musculus* C57BL6/J) | Mouse: C57BL/6J | Jackson laboratory | RRID:IMSR_JAX:000664 | All sexes used |
| Genetic reagent<br>*Mus musculus* C57BL6/J) | *Zcwpw1$^{-/-}$* | PMID:31453335 | | Materials and methods section Generation of Zcwpw1 knockout mice |
| Genetic reagent<br>*Mus musculus* C57BL6/J) | *Zcwpw1$^{KI/KI}$* | This study | | Materials and methods section Mice |
| Genetic reagent<br>*Mus musculus* C57BL6/J) | *Prdm9$^{-/-}$* | This study | | Materials and methods section Mice |
| Antibody | Rabbit polyclonal anti-ZCWPW1 | PMID:31453335 | | IF 1:1000; WB 1:5000; ChIP 35 μg |
| Antibody | Rabbit polyclonal anti-alpha Tubulin | Proteintech Group | 11224–1-AP RRID:AB_2210206 | WB 1:10000 |
| Antibody | Rat polyclonal anti-ZCWPW1 | This study | | Materials and methods section IF 1:200 |

*Continued on next page*

*Continued*

| Reagent type (species) or resource | Designation | Source or reference | Identifiers | Additional information |
|---|---|---|---|---|
| Antibody | Mouse polyclonal anti-SCP3 | Abcam | #ab97672 RRID:AB_10678841 | IF 1:500 |
| Antibody | Rabbit polyclonal anti-SCP1 | Abcam | # ab15090 RRID:AB_301636 | IF 1:2000 |
| Antibody | Rabbit polyclonal anti-RAD51 | Thermo Fisher Scientific | #PA5-27195 RRID:AB_2544671 | IF 1:200 |
| Antibody | Rabbit polyclonal anti-DMC1 | Santa Cruz Biotechnology | #sc-22768 RRID:AB_2277191 | IF 1:100 |
| Antibody | Mouse polyclonal anti-γH2AX | Millipore | #05–636 RRID:AB_309864 | IF 1:300 |
| Antibody | Mouse polyclonal anti-pATM | Sigma-Aldrich | #05–740 RRID:AB_309954 | IF 1:500 |
| Antibody | Rabbit polyclonal anti-MSH4 | Abcam | #ab58666 RRID:AB_881394 | IF 1:500 |
| Antibody | Rabbit polyclonal anti-RNF212 | this study | a gift from Mengcheng Luo, Wuhan University | Materials and methods section IF 1:500 |
| Antibody | mouse polyclonal anti-MLH1 | BD Biosciences | #550838 RRID:AB_2297859 | IF 1:50 |
| Antibody | Rabbit polyclonal anti-H3K4me3 | Abcam | #ab8580 RRID:AB_306649 | IF 1:500 ChIP 5 μg |
| Antibody | Rabbit polyclonal anti-H3K36me3 | Abcam | #ab9050 RRID:AB_30696 | IF 1:500 ChIP 5 μg |
| Antibody | Goat anti-rabbit IgG (H+L) cross-adsorbed secondary Antibody, Alexa Fluor 488 | Thermo Fisher Scientific | #A-11070 RRID:AB_142134 | IF 1:500 |
| Antibody | Goat anti-rabbit IgG H and L (Alexa Fluor 594) preadsorbed | Abcam | ab150084 RRID:AB_2734147 | IF 1:500 |
| Antibody | Goat anti-mouse IgG H and L (Alexa Fluor488) | Abcam | ab150113 RRID:AB_2576208 | IF 1:500 |
| Antibody | Goat anti-mouse IgG H and L (Alexa Fluor 594) preadsorbed | Abcam | ab150120 RRID:AB_2631447 | IF 1:500 |
| Antibody | Goat anti-rat IgG H and L (Alexa Fluor 488) preadsorbed | Abcam | ab150165 RRID:AB_2650997 | IF 1:500 |
| Sequence-based reagent | Primers used in this study | This study | | All primers used in this study are shown in the methods |
| Commercial assay or kit | NEBNext Ultra II DNA Library Prep Kit | NEB | E7645S | |
| Software, algorithm | Trimmomatic v0.32 | doi: 10.1093/bioinformatics/btu170 | http://www.usadellab.org/cms/index.php?page=trimmomatic | |
| Software, algorithm | Bowtie2 v2.3.4.2 | doi: 10.1038/nmeth.1923 | RRID:SCR_005476 | |
| Software, algorithm | Samtools and Picard | doi: 10.1038/ng.806 | http://samtools.sourceforge.net | |
| Software, algorithm | MACS2 v2.1.0 | doi: 10.1186/gb-2008-9-9-r137 | RRID:SCR_013291 | |
| Software, algorithm | Integrative Genomics Viewer | doi: 10.1038/nbt.1754 | RRID:SCR_011793 | |

*Continued on next page*

*Continued*

| Reagent type (species) or resource | Designation | Source or reference | Identifiers | Additional information |
|---|---|---|---|---|
| Software, algorithm | Deeptools2 | doi: 10.1093/nar/gkw257 | http://deeptools.ie-freiburg.mpg.de | |
| Software, algorithm | HOMER software | doi: 10.1016/j.molcel.2010.05.004 | RRID:SCR_010881 | |
| Software, algorithm | GREATER software | doi: 10.1038/nbt.1630 | http://great.stanford.edu | |
| Software, algorithm | regioneR package version1.18.1 | doi: 10.1093/bioinformatics/btv562 | http://www.bioconductor.org/packages/regioneR | |
| Software, algorithm | hisat2 v2.1.0 | doi: 10.1038/nprot.2016.095 | http://github.com/infphilo/hisat2 | |
| Software, algorithm | DESeq2 v 1.18.0 | doi: 10.1186/s13059-014-0550-8 | RRID:SCR_015687 | |
| Software, algorithm | Metacape | doi: 10.1038/s41467-019-09234-6 | http://metascape.org | |

## Mice

The *Zcwpw1* gene (NCBI Reference Sequence: NM_001005426.2) is located on mouse chromosome 5 and comprises 17 exons, with its ATG start codon in exon two and a TAG stop codon in exon 17. The *Zcwpw1* knockout mice were generated in our previous study (*Li et al., 2019a*). The *Zcwpw1* knock-in H3K4me3-reader-dead mutant mice were generated by mutating three sites. The W247I (TGG to ATT) point mutation was introduced into exon 8 in the 5' homology arm, and the E292R (GAG to CGG) and W294P (TGG to CCG) point mutations were introduced into exon 9 in the 3' homology arm. The W247I, E292R, and W294P mutations created in the mouse *Zcwpw1* gene are positionally equivalent to the W256I, E301R, and W303P mutations previously reported in the human *ZCWPW1* gene. To engineer the targeting vector, homology arms were generated by PCR using BAC clones RP24-387B18 and RP24-344E7 from the C57BL/6 library as templates. In the targeting vector, the Neo cassette was flanked by SDA (self-deletion anchor) sites. DTA was used for negative selection. C57BL/6 ES cells were used for gene targeting, and genotyping was performed by PCR amplification of genomic DNA extracted from mouse tails. PCR primers for the *Zcwpw1* Neo deletion were Forward: 5'-CAC TGA GTT AAT CCC ACC TAC GTC-3' and Reverse: 5'CTC TCC CAA ACC ATC TCA AAC ATT-3', with targeted point mutants yielding a 318 bp fragment and WT mice yielding a 174 bp fragment (Cyagen Biosciences Inc, Guangzhou, China).

The mouse *Prdm9* gene (GenBank accession number: NM_144809.3) is located on mouse chromosome 17. Ten exons have been identified, with the ATG start codon in exon one and the TAA stop codon in exon 10. The *Prdm9* knockout mice in the C57BL/6 genetic background were generated by deleting the genomic DNA fragment covering exon 1 to exon nine using the CRISPR/Cas9-mediated genome editing system (Cyagen Biosciences Inc, Guangzhou, China). The founders were genotyped by PCR followed by DNA sequencing analysis. Genotyping was performed by PCR amplification of genomic DNA extracted from mouse tails. PCR primers for the *Prdm9* mutant allele were Forward: 5'-GCT TAG GTA GCA GAA TTG AAG GGA AAG TC-3' and Reverse: 5'- GTT TGT GTC TTT CTA ACT CAA ACT TCT GCA-3', yielding a 580 bp fragment. PCR primers for the *Prdm9* WT allele were Forward: 5'- GCT TAG GTA GCA GAA TTG AAG GGA AAG TC-3' and Reverse: 5'- TCG TGG CGT AAT AAT AGA GTG CCT TG-3', yielding a 401 bp fragment.

All mice were housed under controlled environmental conditions with free access to water and food, and illumination was on between 6 a.m. and 6 p.m. All experimental protocols were approved by the Animal Ethics Committee of the School of Medicine of Shandong University.

## Production of the rat ZCWPW1 antibody

Antibodies to mouse ZCWPW1 were produced by Dia-an Biological Technology Incorporation (Wuhan, China). Briefly, a complementary DNA (cDNA) fragment encoding amino acids 448 to 622 of mouse *Zcwpw1* was inserted into the p-ET-32a + vector (EMD Millipore) and transfected into BL21-CodonPlus (DE3) *Escherichia coli* cells. The cells were cultured at 37℃ overnight and induced

by addition of 0.2 mM isopropyl-1-thio-β-d-galactoside (Sigma-Aldrich) for 4 hr at 28℃. Cells were harvested by centrifugation and disrupted by sonication, and the soluble homogenates were purified by Ni-nitrilotriacetic acid (NI-NTA) Agarose (Qiagen) according to the manufacturer's instructions. The protein was dialyzed in phosphate-buffered saline (PBS) and used to immunize rats, and the anti-serum was affinity-purified on antigen-coupled CNBr-activated agarose (GE Healthcare).

## Tissue collection and histological analysis

Testes from at least three mice for each genotype were dissected immediately after euthanasia, fixed in 4% (mass/vol) paraformaldehyde (Solarbio) for up to 24 hr, stored in 70% ethanol, and embedded in paraffin after dehydration, and 5 µm sections were prepared and mounted on glass slides. After deparaffinization, slides were stained with hematoxylin for histological analysis using an epifluorescence microscope (BX52, Olympus), and images were processed using Photoshop (Adobe).

## Chromosome spread immunofluorescence analysis

Spermatocyte spreads were prepared as previously described (*Peters et al., 1997*). Primary antibodies used for immunofluorescence were as follows: rabbit anti-Zcwpw1 (1:1000 dilution; Dia-an Biological Technology Incorporation [*Li et al., 2019a*]), rat anti-Zcwpw1 (1:200 dilution; Dia-an Biological Technology Incorporation), mouse anti-SCP3 (1:500 dilution; Abcam #ab97672), rabbit anti-SCP1 (1:2000 dilution; Abcam # ab15090), rabbit anti-RAD51 (1:200 dilution; Thermo Fisher Scientific #PA5-27195), rabbit anti-DMC1 (1:100 dilution; Santa Cruz Biotechnology #sc-22768), mouse anti-γH2AX (1:300 dilution; Millipore #05–636), mouse anti-pATM (1:500 dilution; Sigma-Aldrich #05–740), rabbit anti-MSH4 (1:500 dilution; Abcam #ab58666), rabbit anti-RNF212 (1:500 dilution; a gift from Mengcheng Luo, Wuhan University), mouse anti-MLH1 (1:50 dilution; BD Biosciences #550838), rabbit anti-H3K4me3 (1:500 dilution; Abcam #ab8580), and rabbit anti-H3K36me3 (1:500 dilution; Abcam #ab9050). Primary antibodies were detected with Alexa Fluor 488-, 594-, or 647-conjugated secondary antibodies (1:500 dilution, Thermo Fisher Scientific #A-11070, Abcam #ab150084, #ab150067, #ab150113, #ab150120, #ab150119, #ab150165, #ab150168, and #ab150167) for 1 hr at room temperature. The slides were washed with PBS several times and mounted using VECTASHIELD medium with DAPI (Vector Laboratories, #H-1200). Immunolabeled chromosome spreads were imaged by confocal microscopy using a Leica TCS SP5 resonant-scanning confocal microscope. Projection images were then prepared using ImageJ Software (NIH, v. 1.6.0–65) or Bitplane Imaris (v8.1) software.

## Immunoblotting

To prepare protein extracts, tissues were collected from male C57BL/6 mice and lysed in TAP lysis buffer (50 mM HEPES-KOH, pH 7.5, 100 mM KCl, 2 mM EDTA, 10% glycerol, 0.1% NP-40, 10 mM NaF, 0.25 mM Na3VO4 and 50 mM β-glycerolphosphate) plus protease inhibitors (Roche, 04693132001) for 30 min on ice, followed by centrifugation at 4℃ at 13,000 × *g* for 15 min. The supernatants were used for western blotting. Equal amounts of protein were electrophoresed on 10% Bis-Tris protein gels (Invitrogen, NP0315), and the bands were transferred to polyvinylidene fluoride membranes (Millipore). The primary antibodies for immunoblotting included anti-tubulin (1:10,000 dilution; Proteintech Group, #11224–1-AP) and anti-Zcwpw1 (1:5000 dilution; homemade). Immunoreactive bands were detected and analyzed with a Bio-Rad ChemiDoc MP Imaging System and Image Lab Software (Bio-Rad).

## ChIP-seq experiments

PD 14 male mice were used to prepare cells for ChIP-seq (*Chen et al., 2018*). After removal of the tunica albuginea, testes were incubated in 5 ml PBS with collagenase type I (120 U/ml) at 35℃ with gentle agitation for 10 min. The dispersed seminiferous tubules were further digested with 5 ml 0.25% trypsin, plus 0.1 ml DNase I (5 mg/ml) at 35℃ for 8 min, and then terminated by adding 0.5 ml fetal bovine serum (FBS). The resulting suspension was passed through a 70 µm cellular filter. After centrifugation at 500 g for 5 min, the cells were resuspended in PBS and separated into $3 \times 10^5$ per tube. The collected cells from testes were cross-linked in 100 µL of 1% formaldehyde in PBS at room temperature for 10 min and this was followed by 25 µL of 1.25M glycine solution and

mixing via gentle tapping and incubation at room temperature for 5 min. After centrifugation, the cell pellet was washed in PBS three times. Dynabeads Protein A beads (Life Technologies, 10001D) in a total volume of 25 µL were washed twice with 200 µL ice-cold 140 mM RIPA buffer (10 mM Tris-HCl pH 7.5, 140 mM NaCl, 1 mM EDTA, 0.5 mM EGTA, 0.1% SDS, 0.1% Na-deoxycholate, 1% Triton X-100, 1 mM PMSF, 1 × proteinase inhibitor Cocktail, and 20 mM Na-butyrate), followed by resuspension in RIPA buffer to a final volume of 200 µL in a 1.5 ml tube. A total volume of 5 µL H3K4me3 antibody (Abcam, ab8580) or 7 µL Zcwpw1 antibody (homemade, 5 µg/µL) or 5 µL H3K36me3 antibody (Abcam, ab9050) was added to the beads suspension, and this was followed by incubation on a tube rotator for at least 2.5 hr at 4℃. The antibody-coated beads were then washed twice in 140 mM RIPA buffer, followed by resuspension with 200 µL 140 mM RIPA buffer.

The cross-linked cells were incubated in 150 µL lysis buffer (50 mM Tris-HCl pH 8.0, 10 mM EDTA pH8.0, 0.5% SDS, 1 mM PMSF, 1 × proteinase inhibitor cocktail, and 20 mM Na-butyrate) for 20 min on ice, then sonicated using a Diagenode Bioruptor sonication device for 23 cycles (30 s on and then 30 s off). A total volume of 150 µL 300 mM SDS-free RIPA buffer (10 mM Tris-HCl pH 7.5, 300 mM NaCl, 1 mM EDTA, 0.5 mM EGTA, 1% Triton X-100, 0.1% Na-deoxycholate, 1 mM PMSF, 1 × Cocktail proteinase inhibitor, and 20 mM Na-butyrate) and 200 µL 140 mM SDS-free RIPA buffer were added to the samples. After centrifugation at 13,000 × $g$ for 10 min at 4℃, 40 µL supernatant was removed and used as the sample input. The remaining supernatant was transferred to a 1 ml tube containing suspended antibody-coated Protein A beads, and this was followed by incubation on a tube rotator overnight at 4℃.

For the H3K4me3 and H3K36me3 antibodies, the incubated Protein A beads were washed once with RIPA buffer containing 250 mM NaCl, three times with RIPA buffer containing 500 mM NaCl, and once with TE buffer (10 mM Tris-HCl pH 8.0, 1 mM EDTA). For the Zcwpw1 antibody, the incubated Protein A beads were washed twice with RIPA buffer containing 250 mM NaCl, once with RIPA buffer containing 500 mM NaCl, and once with TE buffer. Next, the beads were transferred to a new 0.5 ml tube and incubated in 100 µL ChIP elution buffer (10 mM Tris-HCl pH8.0, 5 mM EDTA, 300 mM NaCl, 0.5% SDS) containing 5 µL proteinase K (Qiagen, 20 mg/ml stock) at 55℃ for 2 hr and then at 65℃ for 4 hr. The eluate was transferred to a 0.5 mL tube, and the enriched DNA was purified by phenol–chloroform, followed by dissolution in 50 µL TE buffer.

An NEBNext Ultra II DNA Library Prep Kit for Illumina (NEB, E7645S) was used for library construction according to the product's instructions. DNA was first end repaired and A-tailed by adding 7 µL NEBNext Ultra II End Prep Reaction Buffer and 3 µL NEBNext Ultra II End Prep Enzyme Mix. Samples were incubated at 20℃ for 30 min and then at 65℃ for 30 min, and finally cooled to 4℃ in a thermal cycler. Adaptor ligation was performed by adding 30 µL NEBNext Ultra II Ligation Master Mix, 1 µL NEBNext Ligation Enhancer, 0.8 µL 200 mM ATP, and 2.5 µL 15 µM Illumina Multiplexing Adaptors. Samples were thoroughly mixed and incubated at 20℃ for 40 min. Following adaptor ligation, 1.2 vol SPRIselect beads (Beckman Coulter, B23318) were used to purify the DNA. PCR amplification was performed with NEBNext Ultra II Q5 Master Mix. The PCR cycle number was evaluated using a FlashGelTM System (Lonza, 57063). The volume of the PCR product was adjusted to 100 µL by adding 50 µL TE buffer. The 300–700 bp DNA fragments were selected with 0.5 volumes plus 0.5 volumes SPRIselect beads and then eluted in 20 µL water. The libraries were sequenced on a Hiseq X-ten instrument set for paired-end 150 bp sequencing (Illumina).

## ChIP-seq bioinformatics analysis

The ChIP-seq raw reads were cropped to 100 bp, and the low quality reads were removed using Trimmomatic v0.32 (*Bolger et al., 2014*). Paired reads were mapped to the mouse genome (version mm10) by Bowtie2 v2.3.4.2 with the parameters '`-X 2000 –no-discordant –no-contain`' (*Langmead and Salzberg, 2012*). Reads with low mapping quality (MAPQ <10) and PCR duplicated reads were removed by Samtools and Picard (*DePristo et al., 2011*; *Li et al., 2009*).

Reads of two replicates were merged to call the necessary peaks, while only one replicate of H3K4me3 and H3K36me3 in Lam et al was used to call peaks in SCP3⁺/H1T⁻ spermatocytes using relatively-stringent conditions. The H3K4me3 peaks in this work and in *Lam et al., 2019* were called by MACS2 v2.1.0 (*Zhang et al., 2008*) with the parameters '`-SPMR -p 0.01 –nomodel`' and '`- SPMR –nomodel -q 0.05`', the Zcwpw1 peaks were called with the parameters '`-SPMR - p 0.001 –nomodel`', the H3K36me3 peaks in Grey et al. and Lam et al. were called with the parameters '`-SPMR –broad -nomodel`' and '`-SPMR –broad -nomodel -p 0.001`', and the

affinity-seq PRDM9 peaks in Walker et al. were called with the parameters '-SPMR -nomodel -p 0.05'. The peaks for DMC1 and PRDM9 in Grey et al. were directly obtained from their published work (*Khil et al., 2012*) and transformed to mm10 using the LiftOver application from UCSC. The SPO11 hotspots in *Lange et al., 2016* were directly obtained from their published work. The peaks intensities were denoted as the fold changes over input lambdas, which were obtained from the results produced by MACS2 callpeak. Zcwpw1 and H3K4me3 (*Lam et al., 2019*) peaks were further selected based on peak intensities greater than a 3-fold enrichment over the input lambda. Affinity-seq PRDM9 peaks in Walker et al. and H3K36me3 peaks in Lam et al. were further selected based on peak intensity greater than a 2-fold enrichment over the input lambda. The normalized signals of H3K4me3, H3K36me3, Zcwpw1, Prdm9, and DMC1 were generated using MACS2 bdgcmp, following the output produced by MACS2 callpeak with SPMR (reads per million for each covered position). The fold change over input lambda worked as the signal enrichment and was transformed into Bigwig using bedGraphToBigWig. ChIP-seq signal tracks were visualized by Integrative Genomics Viewer (*Robinson et al., 2011*). The computeMatrix algorithm in Deeptools2 (*Ramírez et al., 2016*) was used to calculate the normalized signal of each 40bp-size bins in the regions of peak center ±2 k bp. Deeptools plotHeatmap, plotProfile and R (3.4.4) were used to generate the profile plot and heatmap. The script findMotifsGenome.pl function in the HOMER software (*Heinz et al., 2010*) was used to examine the enrichment for transcription factor binding motifs. The gene-region association was determined using the GREATER software (*McLean et al., 2010*). The genomic regions including promoters (TSS ± 2 k bp), exons, introns, intergenic regions, transposon elements, CpG islands, and distal enhancers were downloaded from the UCSC Table Browser under the mm10 version.

## Analysis of the spatial overlap of ZCWPW1 peaks with genomic regions and transposon elements

The peak distribution over genome elements and the overlap between two types of peaks were calculated using bedtools intersect (v2.25.0) with the parameters -u, and minimum overlap was 1 bp (as default). The random binding sites (peaks) used as a control were created with the same number and size distribution as the observed peaks by using the regioneR package version1.18.1 (*Gel et al., 2016*) implemented in R. Using regioneR, a Monte Carlo permutation test with 1000 iterations was performed. In each iteration, the random binding sites were obtained through arbitrarily shuffled in the mouse genome. From this shuffling, the average overlap and standard deviation of the random binding site set was determined, as well as the statistical significance of the association between ZCWPW1 binding sites. Taking the overlap of ZCWPW1 peaks and transposable elements (from RepeatMasker) as an example, pair-end reads were aligned to mouse genome, and then uniquely mapped reads were used to call peaks in the ChIP-seq Bioinformatics Analysis. The random ZCWPW1 peaks were generated using the aforementioned regioneR package. The number of random ZCWPW1 peaks sharing overlap with different types of transposable elements (TE) was obtained in regioneR (whose overlap calculation is same as in bedtoool intersect). The final number of random peaks sharing overlap with transposon elements was indicated as the average of the results of 1000 iterations. To assess the significance of the overlap difference with TEs between observed ZCWPW1 peaks and random peaks, a permutation test was performed in regioneR.

## RNA-seq experiments and bioinformatics analysis

The RNA was extracted from the testis with the Direct-zo RNA MiniPrep kit (Zymo). A total amount of 1.5 µg RNA per sample was used as input material for the RNA sample preparations. Sequencing libraries were generated using the NEBNext Ultra RNA Library Prep Kit for Illumina (NEB, USA) following the manufacturer's recommendations. QC-passed libraries were sequenced on the Hiseq X-ten instrument with the paired-end 150 bp.

The low-quality reads were removed using Trimmomatic v0.32 (*Bolger et al., 2014*). Paired reads were mapped to the mouse genome (version mm10) by hisat2 v2.1.0 (*Pertea et al., 2016*) and to the transcriptome by Salmon v 0.8.2. The DESeq2 v 1.18.0 software (*Love et al., 2014*) was used to identify DEGs from the raw counts produced by Salmon with the two conditions: P adjust < 0.05, and fold change ≥ 2. Metascape (*Zhou et al., 2019*) was used to perform Gene Ontology analysis of DEGs.

## Statistical analysis

Two-tailed Wilcoxon rank sum tests were performed to obtain inferential statistical significance (p values) in related analyses by using the R function wilcox.test. No statistical methods were used to predetermine sample size.

## Acknowledgements

We are grateful for the interesting discussion with K Liu from the University of Hong Kong, China, in the very initial phase of the study. We thank Translational Medicine Core Facility of Shandong University for consultation and instrument availability that supported this work and Jing Xin for assistance with animal work (Shandong University). This work was supported by the Major Program of the National Natural Science Foundation of China [31890780], the National Key Research and Development Programs of China [2018YFC1003400] and the Young Scholars Program of Shandong University (2016WLJH50).

## Additional information

### Funding

| Funder | Grant reference number | Author |
| --- | --- | --- |
| National Key Research and Development Programs of China | 2018YFC1003400 | Hongbin Liu |
| National Natural Science Foundation of China | 31890780 | Hongbin Liu |
| Shandong University | 2016WLJH50 | Hongbin Liu |

The funders had no role in study design, data collection and interpretation, or the decision to submit the work for publication.

### Author contributions

Tao Huang, Conceptualization, Resources, Data curation, Formal analysis, Validation, Investigation, Methodology, Writing - original draft, Writing - review and editing; Shenli Yuan, Data curation, Software, Formal analysis, Investigation, Methodology, Writing - review and editing; Lei Gao, Software, Methodology; Mengjing Li, Xiaochen Yu, Investigation, Writing - original draft; Jianhong Zhan, Chao Liu, Chuanxin Zhang, Gang Lu, Wei Li, Methodology; Yingying Yin, Investigation; Jiang Liu, Conceptualization, Supervision, Writing - review and editing; Zi-Jiang Chen, Hongbin Liu, Conceptualization, Supervision, Funding acquisition, Writing - review and editing

### Author ORCIDs

Tao Huang (iD) https://orcid.org/0000-0002-7086-570X
Wei Li (iD) http://orcid.org/0000-0002-6235-0749
Zi-Jiang Chen (iD) https://orcid.org/0000-0001-6637-6631

### Ethics

Animal experimentation: All mice were housed under controlled environmental conditions with free access to water and food, and illumination was on between 6 am and 6 pm. All experimental protocols were approved by the Animal Ethics Committee of the School of Medicine of Shandong University.

### Decision letter and Author response

Decision letter https://doi.org/10.7554/eLife.53459.sa1
Author response https://doi.org/10.7554/eLife.53459.sa2

# Additional files

## Supplementary files

• Supplementary file 1. Summary of all ChIP-seq experiments indicating antibodies, samples, replicates, genotype and data source.

• Supplementary file 2. ZCWPW1 peaks in WT and *Prdm9* knockout mice.

• Supplementary file 3. Down-regulated and up-regulated genes in *Zcwpw1*$^{-/-}$ (vs WT) whose promoters overlap with ZCWPW1 peaks.

• Transparent reporting form

## Data availability

The raw sequencing data produced in this study (ChIP-seq data listed in Supplementary file 1) and RNA-seq data have been deposited to the Genome Sequence Archive (https://bigd.big.ac.cn/gsa/s/Cjjpbljf) under project PRJCA001901; accession number CRA002088.

The following dataset was generated:

| Author(s) | Year | Dataset title | Dataset URL | Database and Identifier |
|---|---|---|---|---|
| Huang T, Yuan S, Liu J, Chen ZJ, Liu H | 2019 | The histone modification reader ZCWPW1 links histone methylation to repair of PRDM9-induced meiotic double stand breaks | https://bigd.big.ac.cn/gsa/browse/CRA002088 | Genome Sequence Archive, CRA002088 |

The following previously published datasets were used:

| Author(s) | Year | Dataset title | Dataset URL | Database and Identifier |
|---|---|---|---|---|
| Grey C, Clément JA, Buard J, Leblanc B, Gut I, Gut M, Duret L, de-Massy B | 2017 | In vivo binding of PRDM9 reveals interactions with noncanonical genomic sites | http://www.ncbi.nlm.nih.gov/geo/query/acc.cgi?acc=GSE93955 | NCBI Gene Expression Omnibus, GSE93955 |
| Walker M, Billings T, Baker CL | 2015 | Affinity-seq detects genome-wide PRDM9 binding sites and reveals the impact of prior chromatin modifications on mammalian recombination hotspot usage | https://www.ncbi.nlm.nih.gov/geo/query/acc.cgi?acc=GSE61613 | NCBI Gene Expression Omnibus, GSE61613 |
| Lam KG, Brick K, Cheng G, Pratto F, Camerini-Otero RD | 2019 | Cell-type-specific genomics reveals histone modification dynamics in mammalian meiosis | https://www.ncbi.nlm.nih.gov/geo/query/acc.cgi?acc=GSE121760 | NCBI Gene Expression Omnibus, GSE121760 |

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
