## [Decision Letter]

**Acceptance summary:**

This study identifies *Zcwpw1* as an important factor for meiotic DSB repair in mice. Importantly, here the authors generate and analyze a point mutation in *Zcwpw1* that abolishes its binding to histone. *Zcwpw1* localizes to recombination hotspot and is thus clearly a reader of the *Prdm9* deposited histone modifications.

**Decision letter after peer review:**

Thank you for submitting your article "The histone modification reader ZCWPW1 links histone methylation to repair of PRDM9-induced meiotic double stand breaks" for consideration by *eLife*. Your article has been reviewed by three peer reviewers, including Bernard de Massy as the Reviewing Editor and Reviewer #1, and the evaluation has been overseen by Jessica Tyler as the Senior Editor. The reviewers have discussed the reviews with one another and the Reviewing Editor has drafted this decision to help you prepare a revised submission.

Summary:

The authors follow up a prior study where they first described a new gene ZCWPW1. Interestingly, the authors generate a mouse knock in (KI) mutation of the putative methyl-binding region of ZCWPW1. They determine that this mutant displays a similar phenotype to the null with synapsis and DSB repair defects during male meiosis. ChIP-seq analysis identifies numerous peaks of ZCWPW1 binding across chromosomes that correlate with histone methylation marks and PRDM9 binding. Many of these sites appear to depend on PRDM9.

This paper provides some novel information about ZCWPW1 but the presentation of the data is very confusing with references to figures and supplementary figures not ordered in a logical way.

The novel insight from this paper comes from the point mutant analyzed and the description of the chromatin binding of ZCWPW1. However, they are several possible interpretations of these properties and this should be clarified. This approach and data could thus add complementary information with respect to other two papers which among other things reveal the proper DSB localization in the ZCWPW1 KO.

As the authors have submitted their manuscript at the same time as the ones from Mahgoub et al. and Wells et al. , the results from these two papers have also to be integrated in the Discussion to provide a more coherent view or to point out discrepancies if any (i.e.: does ZCWPW1 binds elsewhere than *Prdm9* sites as suggested by Wells et al?). What are the potential implications on the role of ZCWPW1 on gene and TE (transposable elements) expression?

Essential revisions:

1) To show the specificity of the antibody for ZCWPW1 vs. ZCWPW2.

2) Test transcriptome of ZCWPW1 KO, any modification of gene expression and/or TE: analyse transcriptome from 10-12dpp mice and compare with wt and *Prdm*9 KO.

Cellularity of wild type and mutants should be quite close (although not necessarily identical) at these ages, and potential gene expression changes could also be compared with binding sites at promoters.

3) Test in vitro binding affinity of the mutant protein to H3K4, H3K4me3 and H3K36me3.

4) Revise and improve cytology.

5) Analysis of ZCWPW1 binding : explain data, normalization, analyze throughout the genome including TSS, enhancers, CpG islands, along genes.; analyze quantitatively and correlate with other landmarks (H3K4me3, H3K36me3, Spo11, DMC1, PRDM9…), analyse in the PAR (*Prdm9* independent DSB sites), full analysis in *Prdm9* KO, full analysis in ZCWPW1 KI.

6) Strongly revise the whole organisation and presentation of the manuscript and edit for language, remove redundancies, present figures in a proper order as they are cited in the text, make sure all figures and supplementary figures are cited in the text, use a more coherent logic for presenting the ChipSeq localization of ZCWPW1 to hotspots, and for cytology as well, make sure that every conclusion is supported by the data which is presented.

---

## [Author Response]

Summary:The authors follow up a prior study where they first described a new gene ZCWPW1. Interestingly, the authors generate a mouse knock in (KI) mutation of the putative methyl-binding region of ZCWPW1. They determine that this mutant displays a similar phenotype to the null with synapsis and DSB repair defects during male meiosis. ChIP-seq analysis identifies numerous peaks of ZCWPW1 binding across chromosomes that correlate with histone methylation marks and PRDM9 binding. Many of these sites appear to depend on PRDM9.This paper provides some novel information about ZCWPW1 but the presentation of the data is very confusing with references to figures and supplementary figures not ordered in a logical way.

We greatly appreciate the highly positive appraisal of our work by the editors and reviewers. We have now extensively modified our manuscript according to the reviewers’ suggestions.

The novel insight from this paper comes from the point mutant analyzed and the description of the chromatin binding of ZCWPW1. However, they are several possible interpretations of these properties and this should be clarified. This approach and data could thus add complementary information with respect to other two papers which among other things reveal the proper DSB localization in the ZCWPW1 KO.

We have extensively amended our manuscript to include references to the other two manuscripts. Accordingly, the revised text now cites the manuscripts by Mahgoub et al. and Wells et al. (subsection “The H3K4me3 reader function of ZCWPW1 is essential for meiotic recombination”, last paragraph; Discussion, last paragraph).

As the authors have submitted their manuscript at the same time as the ones from Mahgoub et al. and Wells et al. , the results from these two papers have also to be integrated in the Discussion to provide a more coherent view or to point out discrepancies if any (i.e.: does ZCWPW1 binds elsewhere than Prdm9 sites as suggested by Wells et al?). What are the potential implications on the role of ZCWPW1 on gene and TE (transposable elements) expression?

As mentioned above, we now cite the manuscripts by Mahgoub et al. and Wells et al. in all related sections of the revised text (subsection “The H3K4me3 reader function of ZCWPW1 is essential for meiotic recombination”, last paragraph; subsection “ZCWPW1 is an H3K4me3/H3K36me3 reader”, second and last paragraphs; ; Discussion, fourth and last paragraphs).

Essential revisions:1) To show the specificity of the antibody for ZCWPW1 vs. ZCWPW2.

We now have four lines of evidence demonstrating the specificity of the antibody for ZCWPW1 vs. ZCWPW2. For background, recall that there is only 25% (83 aa/331 aa) similarity between ZCWPW1 and ZCWPW2. The first line of evidence is as follows: we raised the antibody against a 175 amino acid peptide unique to the C-terminal region of ZCWPW1. There is only 8.8% (29 aa/331 aa) similarity between the C-terminal regions of ZCWPW1 and ZCWPW2. Epitopes are peptides generally longer than 5 amino acids, and we did not find the same potential epitopes between ZCWPW1 and ZCWPW2. Second, our western blots established that the antibody we produced recognizes a specific band of the expected size in the wild type but not the *Zcwpw1^-/-^* testes (Figure 1—figure supplement 2A). Third, immunofluorescence staining shows that ZCWPW1 mainly localizes to WT nuclei of leptotene and zygotene spermatocytes but disappears in *Zcwpw1^−/−^*spermatocytes (Figure 1—figure supplement 2B). Fourth, our ChIP-seq shows that ZCWPW1 can specifically bind to histone modification sites deposited by PRDM9, while there are no binding peaks in *Zcwpw1^−/−^* or *Zcwpw1^KI/KI^* mice (Figure 3A, D). Thus, we have four separate lines of empirical evidence supporting the specificity of our antibody for ZCWPW1 vs. ZCWPW2. We appreciate the necessity of establishing this point to justify our specific claims in this study, and we thank the reviewers for focusing our attention here.

2) Test transcriptome of ZCWPW1 KO, any modification of gene expression and/or TE: analyse transcriptome from 10-12dpp mice and compare with wt and Prdm9 KO.Cellularity of wild type and mutants should be quite close (although not necessarily identical) at these ages, and potential gene expression changes could also be compared with binding sites at promoters.

In the revised version, we have written the text as the follows:

“Because we found that 1,766 of the ZCWPW1 binding sites overlapped with promoters, we sought to examine the transcriptome in *Zcwpw1^−/−^* testes by RNA-seq to investigate whether ZCWPW1 affected the expression of those genes whose promoters overlapped ZCWPW1 binding sites. […] These data strongly suggest that ZCWPW1 might not affect the transcription level of genes even though it can bind to their promoter regions.”

3) Test in vitro binding affinity of the mutant protein to H3K4, H3K4me3 and H3K36me3.

We thank the reviewer for raising this very important question which, but unfortunately this cannot be addressed with certainty at present. To achieve social distancing in the face of the COVID19 pandemic, our laboratory has been closed since 23 January 2020. However, several other pieces of evidence can potentially offer the same support. For example, Mahgoub et al. confirmed that recombinant ZCWPW1 (1–440aa) binds with the highest affinity to H3K4me3/K36me3 peptides in vitro (Mahgoub et al., 2019). In addition, He et al. have confirmed that the human zf-CW domain preferentially binds to the histone H3 tail with trimethylated K4, and in contrast the zf-CW mutant (W256I/E301R/T302L/W303P) lost its binding to H3K4me3 and H3K4me0 (He et al., 2010). Most importantly, we have also performed an additional ZCWPW1 ChIP-seq analysis of testes samples from PD14 (postnatal day 14) WT, *Zcwpw1^−/−^*, and *Zcwpw1^KI/KI^* mice. This analysis clearly indicated that no ZCWPW1 peaks were detected in the *Zcwpw1^−/−^* or *Zcwpw1^KI/KI^* mice (Figure 1A and D). These in vivo results, viewed alongside the previous reports of ZCWPW1 function in the meiotic process in addition to reports demonstrating that these specific mutations in ZCWPW1's zf-CW domain affect the protein's ability to read histone modifications (including H3K4me3), together support our conclusion that the ZCWPW1^W247I/E292R/W294P^ variant is an H3K4me3-reader-dead variant of ZCWPW1. We hypothesize that the mutant protein is not only an H3K4me3-reader-dead variant of ZCWPW1, but is also a histone binding-dead variant.

4) Revise and improve cytology.

We appreciate this very constructive critique from the reviewer, and we have worked hard to improve our cytology reporting. First, we have re-organized all of the cytology for WT, *Zcwpw1^−/−^*, and *Zcwpw1^KI/KI^* mice. Now, the Results section begins directly with the phenotypic analysis of WT, *Zcwpw1^−/−^*, and *Zcwpw1^KI/KI^* mice (Figure 1 and 2) as the basis for exploration of the underlying mechanism of ZCWPW1 involvement in meiosis prophase I. Second, we have quantified the number of synapsis chromosomes in WT, *Zcwpw1^−/−^*, and *Zcwpw1^KI/KI^* spermatocytes to show the synapsis defects in mutant mice (Figure 1—figure supplement 2C). In addition, we have quantified the number of foci for DMC1, RAD51, MSH4, and RNF212 in WT and *Zcwpw1^KI/KI^* spermatocytes to show the recombination defects in mutant mice (Figure 1D and F; Figure 1—figure supplement 3B and D).

In addition, it bears mentioning that we have tested several commercial and homemade antibodies against MSH4 and RNF212 for immunofluorescence staining of spermatocytes, most of which did not work. We found that one commercial MSH4 antibody (Abcam #ab58666 Lot:GR319120-8) and a homemade RNF212 antibody (a gift from Mengcheng Luo, Wuhan University) both worked but with some background noise. We also designed a new immunization strategy to generate new MSH4 and RNF212 antibodies in the hope that they can perform well in spermatocyte immunofluorescence.

5) Analysis of ZCWPW1 binding : explain data, normalization, analyze throughout the genome including TSS, enhancers, CpG islands, along genes.; analyze quantitatively and correlate with other landmarks (H3K4me3, H3K36me3, Spo11, DMC1, PRDM9…), analyse in the PAR (Prdm9 independent DSB sites), full analysis in Prdm9 KO, full analysis in ZCWPW1 KI.

We thank the reviewer for suggesting this analysis which has absolutely led to the overall improvement of our study and our manuscript. We have made a major effort to address the remaining concerns, detailed below.

All of the ChIP-seq signals in this study were normalized, as described in the Materials and methods. Specifically, we first generated the whole genome signal for each base with RPM (reads per million reads). Then, we calculated the fold change over input as the final normalized signal for the whole genome. For the signal in specific regions, we calculated the normalized signal for 40 bp bins using computeMatrix in the Deeptools2 software. We have subsequently fully analyzed ZCWPW1 binding throughout the genome in the revised manuscript (subsection “ZCWPW1 is an H3K4me3/H3K36me3 reader”).

The quantification and correlation analysis between ZCWPW1 and other landmarks (i.e., H3K4me3, H3K36me3, PRDM9, DMC1, SPO11) are provided in the revised text (subsections “ZCWPW1 is an H3K4me3/H3K36me3 reader”; “*Zcwpw1* binding is strongly promoted by the histone modification activity of *Prdm9*”; “*Zcwpw1* localizes to DMC1-labelled DSB hotspots in a PRDM9-dependent manner”). In addition, we have fully analyzed the ZCWPW1 binding in *Prdm9^−/−^*mice (subsections “*Zcwpw1* binding is strongly promoted by the histone modification activity of *Prdm9*”; “*Zcwpw1* localizes to DMC1-labelled DSB hotspots in a PRDM9-dependent manner”). We conducted an additional ZCWPW1 ChIP-seq analysis of testes samples from PD14 WT, *Zcwpw1^−/−^*, and *Zcwpw1^KI/KI^* mice. This analysis clearly indicated that no ZCWPW1 peaks were detected in the *Zcwpw1^−/−^* or *Zcwpw1^KI/KI^* mice (Figure 3A and D). We also found that ZCWPW1 could localize to DMC1-labelled hotspots in the pseudo-autosomal region similarly to its localization in autosomes.

6) Strongly revise the whole organisation and presentation of the manuscript and edit for language, remove redundancies, present figures in a proper order as they are cited in the text, make sure all figures and supplementary figures are cited in the text, use a more coherent logic for presenting the ChipSeq localization of ZCWPW1 to hotspots, and for cytology as well, make sure that every conclusion is supported by the data which is presented.

We are very grateful to the reviewer’s constructive comments and suggestions that have helped us to improve the coherence of our manuscript. We have modified the whole manuscript to accommodate the reviewer’s suggestions. Specifically, we would also like to note that the revised manuscript has been professionally edited for grammar and usage by a PhD-level scientific editor with a background in molecular biology.